

# Characterization and first results from LACIS-T: A moist-air wind tunnel to study aerosol-cloud-turbulence interactions

Dennis Niedermeier[1], Jens Voigtländer[1], Silvio Schmalfuß[1], Daniel Busch[1], Jörg Schumacher[2], Raymond A. Shaw[3], and Frank Stratmann[1]

[1]Department of Experimental Aerosol and Cloud Microphysics, Leibniz Institute for Tropospheric Research, Leipzig, Germany
[2]Department of Mechanical Engineering, Technische Universität Ilmenau, Ilmenau, Germany
[3]Department of Physics, Michigan Technological University, Houghton, USA

**Correspondence:** D. Niedermeier (niederm@tropos.de)

**Abstract.** The interactions between turbulence and cloud microphysical processes have been investigated primarily through numerical simulation and field measurements over the last ten years. However, only in the laboratory we can be confident in our knowledge of initial and boundary conditions, and are able to measure under statistically stationary and repeatable conditions. In the scope of this paper, we present an unique turbulent moist-air wind tunnel, called the Turbulent Leipzig Aerosol Cloud

Interaction Simulator (LACIS-T) which has been developed at TROPOS in order to study cloud physical processes in general and interactions between turbulence and cloud microphysical processes in particular. The investigations take place under well-defined and reproducible turbulent and thermodynamic conditions covering the temperature range of warm, mixed-phase and cold clouds ($25°C > T > -40°C$). The continuous-flow design of the facility allows for the investigation of processes occurring on small time (up to a few seconds) and spatial scales (micrometer to meter scale) and with a Lagrangian perspec-

tive. The experimental studies using LACIS-T are accompanied and complemented by Computational Fluid Dynamics (CFD) simulations which help us to design experiments as well as to interpret experimental results.

In this paper, we will present the fundamental operating principle of LACIS-T, the numerical model as well as results concerning the thermodynamic and flow conditions prevailing inside the wind tunnel combining both characterization measurements and numerical simulations. Finally, first results are depicted from deliquescence / hygroscopic growth as well as droplet

activation and growth experiments. We observe clear indications of the effect of turbulence on the investigated microphysical processes.

## 1 Introduction

Clouds are important players in both weather and climate. They are the source of precipitation and significantly contribute to the radiative budget of the Earth. Extensive research activity during the last decades has been carried out in order to understand

cloud processes and related interactions. As a consequence, the quantitative knowledge has increased tremendously in the past decades but many of the occurring interactions and their influence on weather and climate are still poorly understood and ill quantified (Quaas et al., 2009; Seinfeld et al., 2016).





Atmospheric clouds are often non–stationary, inhomogeneous, intermittent, and cover an enormous range of spatial (micrometers to hundreds of kilometers) and temporal (microseconds to hours and days) scales. Cross–scale interactions between turbulent fluid dynamics and cloud microphysical processes influence cloud behavior and cloud development (Bodenschatz et al., 2010). Turbulence drives processes such as entrainment and mixing, leading to strong fluctuations in aerosol particle concentration, temperature, water vapor, and consequently supersaturation which affects cloud droplet activation, growth and decay (Siebert et al., 2006). It links to phase transition processes of water as well as particle collisions and breakup (Shaw, 2003). These processes, in turn, can have buoyancy and drag effects on turbulence and influence cloud dynamic processes up to the largest scales (Stevens et al., 2005; Malinowski et al., 2008; Bodenschatz et al., 2010).

The remote location (high above the ground) and transience of clouds makes comprehensive characterization of clouds and their environment very difficult. Moreover, the intermittent nature of clouds requires the observation of a large number of clouds before statistics will converge. The study of atmospheric clouds is therefore an ambitious, expensive and technically challenging undertaking. In order to better understand and quantify the behavior of clouds in general, and the interactions between turbulence and cloud microphysical processes in particular, atmospheric observations alone are far from sufficient, and intensive laboratory investigations under well-defined and reproducible conditions form an irreplaceable part of cloud research (Stratmann et al., 2009).

A number of laboratory facilities for aerosol and cloud research, such as aerosol–cloud chambers, continuous flow systems, wind tunnels, and electrodynamic balances have been developed and used over the last decades for atmospheric research (a detailed compilation of, and references for, atmospheric chambers and facilities is given in Chang et al. (2016)). At TROPOS, we developed the laminar flow tube LACIS (Leipzig Aerosol Cloud Interaction Simulator, Stratmann et al., 2004; Hartmann et al., 2011) which has been applied for the investigation of aerosol–cloud interaction processes under controllable and reproducible conditions in a continuous-flow setting. Investigations using LACIS comprised the consistent descriptions of both hygroscopic growth and droplet activation for various inorganic (Wex et al., 2005, 2006; Niedermeier et al., 2008) and organic materials such as HULIS (HUmic LIke Substances, Wex et al., 2007; Ziese et al., 2008), secondary organic aerosol (Wex et al., 2009; Petters et al., 2009) and soot particles (Henning et al., 2010; Stratmann et al., 2010). Further, LACIS has been used for the investigation and quantification of the immersion freezing behavior of various mineral dust (Niedermeier et al., 2010, 2011, 2015; Augustin-Bauditz et al., 2014; Wex et al., 2014; Hartmann et al., 2016), ash (Grawe et al., 2016, 2018) and biological particles (Augustin et al., 2013; Hartmann et al., 2013).

These results and those of the other laboratory chambers and facilities have been fundamental in filling gaps in the big puzzle of understanding aerosol–cloud interactions. However, the investigations at LACIS and many of those at other facilities were carried out having average and/or slowly changing thermodynamic conditions in the vicinity of the particle/droplet, i.e., neglecting possible influences of turbulent fluctuations in these properties. Only very few experimental set-ups are available so far for laboratory investigations of aerosol–cloud–turbulence interactions due to the demanding experimental requirements regarding the accuracy and reproducibility of the experimental parameters (e.g. temperature, humidity, particle properties, turbulence parameters). One example is the Pi chamber which is a turbulent aerosol–cloud reaction chamber studying cloud processes on time scales of minutes to hours (Chang et al., 2016).


In the scope of this paper, we introduce a turbulent moist-air wind tunnel, called LACIS-T (Turbulent Leipzig Aerosol Cloud Interaction Simulator), which has been developed at TROPOS in order to study cloud physical processes in general

and interactions between turbulence and cloud microphysical processes, such as droplet / ice crystal formation, in particular. The investigations take place under well-characterized and reproducible turbulent and thermodynamic conditions covering the temperature range of warm, mixed-phase and cold clouds ($25°C > T > -40°C$). The continuous-flow design of the facility allows for the investigation of processes occurring on small time (up to a few seconds) and spatial scales (micrometer to meter scale) and with a Lagrangian perspective, in contrast to other facilities like the Pi chamber. A specific benefit of LACIS-T is

the well-defined location of aerosol particle injection directly into the turbulent mixing zone as well as the precise control of the respective initial / boundary flow velocity and thermodynamic conditions. The size and number concentration of aerosol and cloud particles can be measured at different defined locations below the aerosol injection.

The experimental studies using LACIS-T on aerosol–cloud–turbulence interactions are accompanied and complemented by Computational Fluid Dynamics (CFD) simulations which help us to design experiments, i.e., obtain suitable experimental

parameters, as well as to interpret experimental results. The simulations are performed in OpenFoam® for modeling flow, heat and mass transfer as well as particle and droplet dynamics. In that context we formulated e.g., an Eulerian–Lagrange approach so that the growth of individual cloud particles can be tracked along their trajectories through the simulation domain (see e.g. Kumar et al., 2018).

In the following, LACIS-T and the currently available instrumentation will be described (Sect. 2). The numerical model and

boundary conditions as well as the numerical particle tracking method are explained in detail in Sect. 3. Afterwards, we present results concerning the thermodynamic and flow conditions prevailing in LACIS-T combining both characterization measurements and numerical simulations (Sect. 4). First results from deliquescence / hygroscopic growth as well as droplet activation and growth experiments are depicted in Sect. 5. Finally, we will close with a summary as well as an outlook concerning cloud microphysical processes we will address with LACIS-T in the near future.

## 2   Technical Description of LACIS-T

The objective of the wind tunnel design is to generate a locally-homogeneous and isotropic turbulent air flow into which aerosol particles can be injected, and in which the water vapor saturation is precisely controlled. Under suitable conditions, aerosol particles act as cloud condensation nuclei (CCN) or ice-nulceating particles (INP) and cloud droplet formation or heterogeneous ice nucleation and subsequent growth within a turbulent environment is observed. The turbulent flow is created

as air flows past passive grids, as described in more detail later in this section. The primary novelty of the wind tunnel is the existence of two parallel paths through which air flows and is humidified, which are combined in the turbulent flow region. Because of the nonlinearity of the saturation water vapor pressure on temperature (Clausius–Clapeyron equation), the resulting mixture can be supersaturated. Specifically, supersaturated environment is created through the generally known process of isobaric mixing (Bohren and Albrecht, 1998). The exact humidity within the turbulent region depends on the temperatures and

humidities within the two streams, as well as the location within the turbulent mixing layer in the wind tunnel.





LACIS-T is a closed-loop wind tunnel (Göttingen type) which has been designed and built up at TROPOS in collaboration with engineering offices "Ingenieurbüro Dr.-Ing. W. Lorenz-Meyer", and "Ingenieurbüro Mathias Lippold, VDI, Windkanalkonstruktion und Windkanaltechnik". A schematic of the construction is shown in Fig. 1. The main components are radial blowers, particle filters, valves, flow meters, the humidification system, heat exchangers, turbulence grids, the measurement

section and the adsorption dehumidifying system. These components are applied in order to generate the two particle-free air flows each of which is conditioned to a certain temperature and water vapor concentration. These two conditioned particle-free air flows are turbulently mixed inside the measurement section and aerosol particles are injected into the mixing zone of the two particle-free air flows enabling studies of aerosol–cloud–turbulence interaction performed at ambient pressure. The mean velocity inside the measurement section can be varied between 0.5 and 2 m/s. The detailed description of LACIS-T's design

and its functionality will be given in the following, including the adjustment procedure in terms of the thermodynamic — mainly temperature and water vapor concentration — and flow conditions.

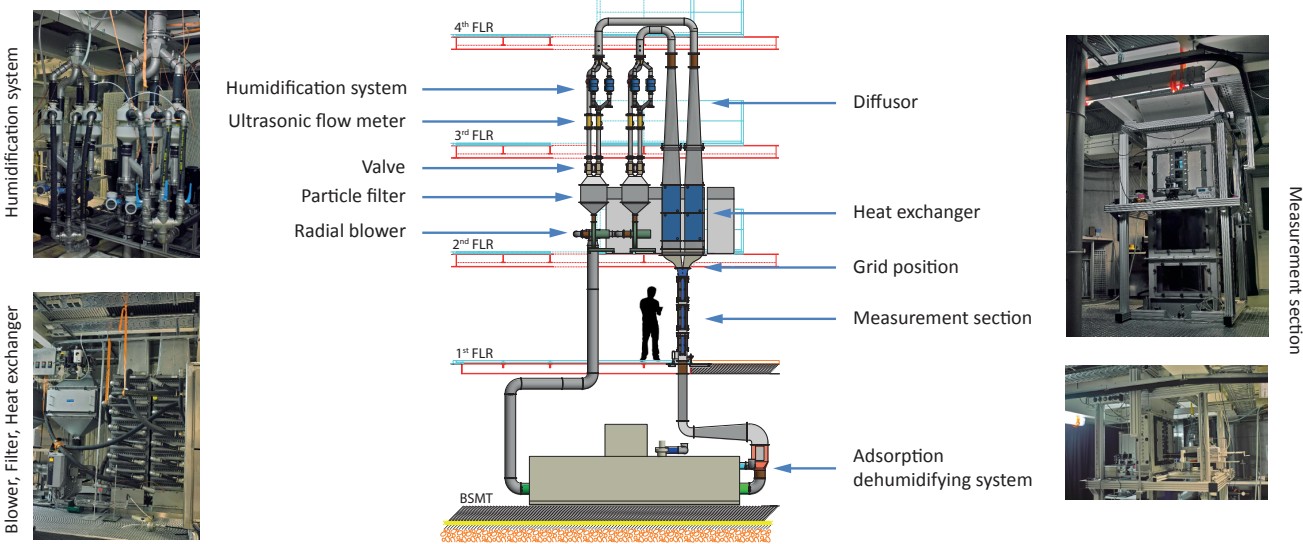

**Figure 1.** A schematic of LACIS-T including photos of individual components (© by Ingenieurbüro Mathias Lippold, VDI; TROPOS).

Two radial blowers (Witt & Sohn AG, Germany) separately drive the two particle-free, dry air flows (flow branches 'A' and 'B'). Flow rates of up to 6.000 l/min in each flow branch are possible. Afterwards, each flow passes a particle filter (Filter class U16; TROX GmbH, Germany) to remove aerosol particles. Subsequently, a defined amount of water vapor can be added to

each of the particle-free air flows by means of a humidification system. For each flow branch, it consists of three humidifiers made of Nafion (FC600-7000, Permapure Inc., USA) being surrounded by water jackets and a bypass where the air remains unhumidified. A pump (Grundfos GmbH, Germany) circulates water through the three Nafion humidifiers (150 l/min each), and a reservoir compensates the water loss due to water vapor transport through the Nafion tubes humidifying the particle-free air





flow. Heat exchangers, being connected with thermostats (Huber unistat 510, Peter Huber Kältemaschinenbau AG, Germany),
keep the temperature of the water-jackets at a defined value. The humidifiers are used in a counter flow fashion. Pneumatically
driven valves (Valtek MaxFlo 3, Flowserve Essen GmbH, Germany) are used to adjust the respective volume flows through
the bypass and the humidifiers and ultrasonic flow meters (Prosonic Flow, Endress+Hauser AG, Switzerland) are applied to
measure the respective volume flow rate. Due to the setup it is possible to a) ensure measurements under dry conditions by-
passing the humidifiers and b) to mix dry and humidified air in order to reach dew-point temperatures below 0°C since the
functionality of the Nafion tubes is limited to a temperature range above the melting point of ice. The dew point temperature of
each branch is monitored downstream of the humidification system by means of dew point mirrors (DPM, MBW 973, MBW
Calibration, Switzerland), which feature an accuracy of 0.1 K.

Then, the two air streams are re-directed and passed through diffusers, changing the cross section from circular to rectan-
gular. This is needed for the entrance into the heat exchangers (Wätas Wärmetauscher Sachsen GmbH, Germany). Each heat
exchanger contains a coolant, the temperature of which can be adjusted to a defined temperature using thermostats (Huber
unistat 915, Peter Huber Kältemaschinenbau AG, Germany). The temperature can be adjusted between $-40$°C and $25$°C. The
dimension of the heat exchangers has been chosen to be able to cool the air flows down to $-40$°C without any significant
undercooling of the cooling liquid which would lead to condensation/deposition of water vapor at the inner walls and therefore
to a loss of water vapor.

Downstream of the heat exchangers, both particle-free air flows are precisely conditioned in terms of volume flow rate,
water vapor content and temperature. Before entering the measurement section, the air flows pass passive square-mesh grids
(mesh length of $M = 1.9$ cm, rod diameter $d_{\mathrm{rod}} = 0.4$ cm and a blockage of $\sigma_b = 30\%$) which are situated $20$ cm above the
measurement section (see Fig. 2). This configuration has been chosen to create turbulence that is approximately isotropic in
the center-region of the measurement section and is homogeneous in transverse planes.

At the inlet of the measurement section, the two conditioned particle-free air flows are merged and turbulently mixed. A
wedge-shaped "cutting edge" separates both air flows right above the inlet of the measurement section (see right picture in
Fig. 2). Three rectangular feed-throughs, which represent the aerosol inlet ($20$ mm x $1$ mm each, $1$ mm separation between
feed-throughs) are located in the center of this cutting edge. Here, the aerosol flow is introduced into the mixing zone of the
two particle-free air flows. Size-selected, quasi monodisperse aerosol particles of known chemical composition can be injected.
Size selection is conducted via a DMPS-System (Differential Mobility Particle Sizer) which includes a DMA (Differential
Mobility Analyzer, Knutson and Whitby (1975), type "Vienna medium") for selecting a narrow dry particle size fraction and a
CPC (Condensational Particle Counter, TSI 3010, TSI Inc, USA) to obtain the particle concentration. The particles themselves,
which can serve as CCN and/or INP, are generated by means of an Atomizer (TSI 3075, TSI Inc., USA) or a Fluidized Bed
Generator (TSI 3400A, TSI Inc., USA).

The measurement section itself is a rectangular prism and $200$ cm long, $80$ cm wide, and $20$ cm deep (Fig. 2). It features
various instruments for characterizing the prevailing thermodynamic, turbulence, and microphysical properties. This includes
measurements of temperature, mean water vapor concentration, flow velocity, turbulence intensity, and dissipation rate as well
as cloud particle size distributions at various locations. A summary of the instrumentation available so far is given in Table 1.





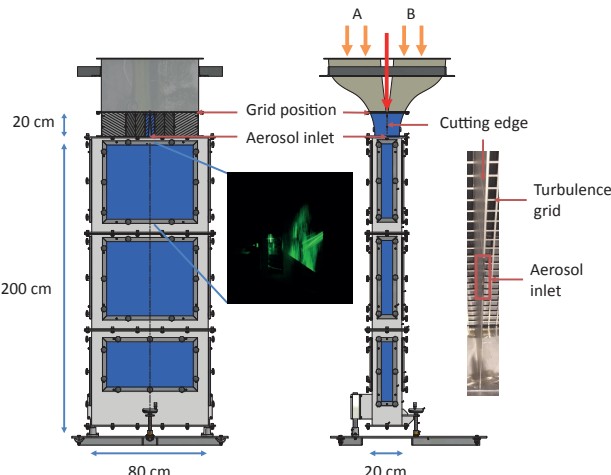

**Figure 2.** A sketch of the measurement section is shown including its dimensions as well as the position of the turbulence grids, the cutting edge and the aerosol inlet (© by Ingenieurbüro Mathias Lippold, VDI; TROPOS). The red arrow marks the location where the particles are injected. The picture in between the sketches of the measurement section shows a formed cloud which is illuminated by a green laser light sheet.

The design of the measurement section ensures flexibility in terms of instrument mounting. That means the windows shown
in the schematic drawing of the measurement section in Fig. 2 can be replaced by panels with defined positions for access ports as well as customized optical windows. Please note that the measurement section is currently not heat insulated. However, as shown in Sect. 4, wall effects have a negligible influence on the processes occurring in the mixing zone for the experiments carried out so far.

After passing through the measurement section, the entire flow is dried and heated by means of an adsorption dehumidifying
system (Marquardt & Schaupp Luftentfeuchtungssysteme GmbH, Germany). Then the flow splits up again into two branches and the whole cycle starts again.

In summary, we are able to separately adjust volume flow rate, temperature and dew-point temperature of each flow branch so that different experimental configurations are possible. This is an important requirement for the experimental studies performed at LACIS-T especially with regard to our first investigations about deliquescence / hygroscopic growth as well as droplet
activation and growth experiments which will be presented in Sect. 5. Two different settings have been applied for these experiments which will be briefly introduced in the following. In the first, isothermal setting, both particle-free air flows feature the same conditions in terms of flow rate, temperature and dew-point temperature. The temperature and dew-point temperature of the aerosol flow are adjusted independently of the two particle-free air flows such that microphysical processes like deliquescence and hygroscopic growth of aerosol particles can be studied. In the second, non-isothermal setting, the two
particle-free air flows are differently conditioned in terms of temperature and dew-point temperature, i.e., there is a temperature





**Table 1.** Available instrumentation for the generation of defined aerosol particles, for the determination of the respective flow, air and dew-point temperature as well as for measuring cloud particle sizes and numbers inside the measurement section.

| Instrumentation | Application |
| --- | --- |
| Atomizer | Generation of aerosol particles from a solution or suspension |
| Fluidized Bed Generator (TSI 3400A) | Generation of aerosol particles from a dry reservoir |
| Differential Mobility Particle Sizer (DMPS) | Generation and counting of size-selected monodisperse aerosol particles |
| Hot-wire anemometer (Dantec Dynamics Inc.) | Measurement of flow velocity (mean and fluctuations) at various locations |
| Cold-wire anemometer (Dantec Dynamics Inc.) | Measurement of temperature fluctuations at various locations |
| Pt100 resistance thermometers | Measurement of mean temperature at various locations |
| Dew-point mirror (DPM, MBW 973, MBW Calibration) | Measurement of mean dew-point temperature at various locations |
| Promo 2000 with welas 2300 aerosol spectrometer (PALAS GmbH) | Determination of cloud particle size and number at various locations |

difference $\Delta T$ between both air-flows. If the relative humidities in both air flows are sufficiently high, supersaturation is achieved when both flows are mixed.

## 3 Numerical simulations

As mentioned above, measurements of flow, thermodynamic and cloud particle properties are performed at various locations

inside the measurement section. However, it is challenging to determine a comprehensive picture of, for example, the instantaneous parameter fields. Therefore, the experimental investigations are accompanied and complemented by CFD simulations performed in OpenFOAM®. The simulations will further be very helpful for the design of experiments, i.e., obtaining suitable experimental parameters, as well as for the interpretation of experimental results. The simulations will also aid in development of physically-sound parameterizations concerning aerosol–cloud–turbulence interaction. In the following the numerical setup

for flow and particle dynamics simulations will be presented.





### 3.1 Computational domain and numerical grid

As already shown, the measurement section is rectangular prism being 200 cm long, 80 cm wide, and 20 cm deep. For the simulations, the computational domain comprises only a part of the wind tunnel upstream the aerosol injection, including the turbulence grid in order to reduce computational effort. It covers the upper 80 cm of the measurement section, and it is 11.5 cm

wide and 20 cm deep (Fig. 3). This is the region of interest for the measurements carried out so far. This domain is decomposed into a grid with approximately $7.6 \times 10^6$ cells. Multi region support, i.e., a coupled simulation of solid and fluid regions, is necessary for non-isothermal setups, as the region between the walls above the aerosol inlet is a solid and thus conducts heat, so that the wall temperature is not fixed, but actually depends on the behavior of fluid and solid temperature at the walls. The solid region was decomposed into approximately 30,000 grid cells.

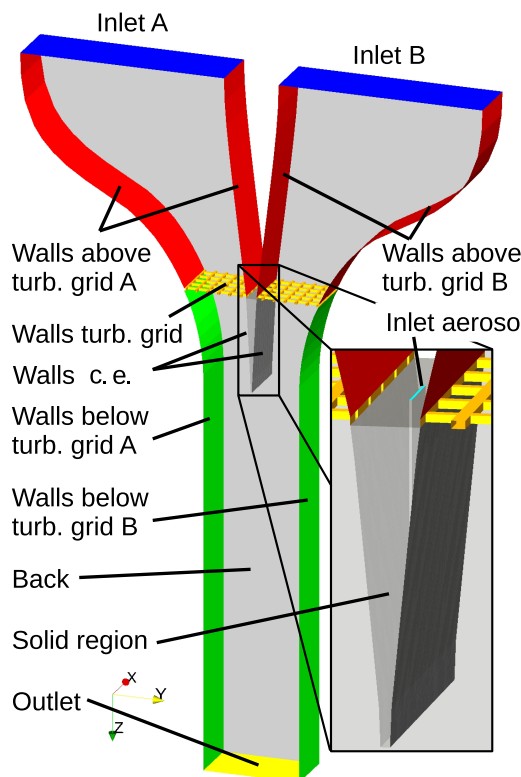

**Figure 3.** Boundaries of the computational domain with detailed view of the aerosol inlet region between the two air flow branches (c.e. stands for cutting edge).

### 180 3.2 Fluid flow and heat / mass transfer simulations

For an isothermal (i.e., the temperature in both flow branches A and B is identical, $T_A = T_B$), unhumidified (i.e., dry) setup, as will be described in Sect. 4.1, OpenFOAM®'s solver "pimpleFoam" is used, which is a finite volume solver for incom-





**Table 2.** Boundary conditions for LES.

| | $\bar{u}\,[m/s]$ | $q_\mathrm{v}$ | $T$ | $p$ [Pa] |
|---|---|---|---|---|
| Inlet A | (0 0 0.35) | $q_{\mathrm{v,A}}$ | $T_\mathrm{A}$ | zero gradient |
| Inlet B | (0 0 0.35) | $q_{\mathrm{v,B}}$ | $T_\mathrm{B}$ | zero gradient |
| Outlet | zero gradient | zero gradient | zero gradient | 101325 |
| Walls cutting edge | no slip | zero gradient[1] | coupled[2] | zero gradient |
| Walls turb. grid | no slip | zero gradient[1] | zero gradient | zero gradient |
| Walls above turb. grid A | no slip | zero gradient[1] | fixed value[3] | zero gradient |
| Walls above turb. grid B | no slip | zero gradient[1] | fixed value[3] | zero gradient |
| Front/back | cyclic | cyclic | cyclic | cyclic |
| Walls below turb. grid A | no slip | zero gradient[1] | fixed value[3] | zero gradient |
| Walls below turb. grid B | no slip | zero gradient[1] | fixed value[3] | zero gradient |
| Inlet aerosol | (0 0 1.25) | $q_{\mathrm{v,I}}$ | fixed value[3] | zero gradient |

[1] If the value of the water vapor mixing ratio $q_v$ exceeds the value which corresponds to a relative humidity of 100%, it is set to a fixed value boundary condition, so that RH stays at a maximum of 100% at the walls. [2] The region between the left and right side of the cutting edge is simultaneously simulated as a heat conducting solid and the temperatures at the wall are the result of this simulation. [3] Values according to measurement.

pressible, transient, turbulent flows. Regarding non-isothermal ($T_\mathrm{A} \neq T_\mathrm{B}$) and humidified flows, we use an adapted version of OpenFOAM's "chtMultiRegionFoam", which is able to simulate multi region heat transport and which was extended to also

transport the mass fraction of water vapor.

As turbulent fluctuations are to be investigated, it is not suitable to use Reynolds Averaged Navier Stokes (RANS) models. This would only provide mean values of turbulent properties. A fully resolved simulation (Direct Numerical Simulation, DNS) is far beyond available computational resources, but is planned as a possible future work. So the method of choice so far is a Large Eddy Simulation (LES), calculating the larger, energy containing eddies and modeling the smallest eddies, so it is not as

computationally expensive as a DNS. We choose the dynamic k-equation LES model, as it has proven to be a good model for decaying turbulence and the transport of thermodynamic quantities (Chai and Mahesh, 2012).

The air inside LACIS-T is considered to be an ideal gas with a molar mass of 28.97 g/mol, a heat capacity of 1.007 kJ/(kg K), and a dynamic viscosity of $18.23 \cdot 10^{-6}$ Pa s. Regarding the solid region, approximate properties of steel are used with a molar mass of 56 g/mol, a thermal conductivity of 40 W/(m K), a specific heat capacity of 500 J/(kg K), and a density of 8000 kg/m³.

Gravitational acceleration is set to 9.81 m/s² in the vertical direction (positive z-direction in Fig. 3).

The different boundary conditions are depicted in Fig. 3 and listed in detail in Table 2. At the inlets, fixed values of velocity, temperature, and humidity were set according to measurements (inlet A and inlet B) or set values, respectively. The pressure gradient is set to zero.

As usual, a no-slip condition for the velocity is assumed at all walls. Wall temperatures have been measured above and

below the turbulence grid and the according walls are set to the appropriate values. The turbulence grid is assumed to have





approximately the same temperature as the surrounding fluid and thus the temperature gradient is set to zero there. For the humidity, we use a mixed wall boundary condition. For saturated flow conditions, the relative humidity at the wall is set to 100%. The condensed water is lost and heat release due to condensation is neglected. If the relative humidity at a wall is below 100%, the gradient is assumed to be zero.

At the outlet, zero gradient conditions are used for velocity, temperature, and humidity. In other words, the respective partial derivate vanishes in normal direction to the outlet surface. To reduce the computational effort, only a 11.5 cm wide section of LACIS-T was modeled, as mentioned above. At the front and back of this section, periodic boundary conditions are used.

The simulations are first run for 1 s of physical time, starting with a steady state solution, until a quasi steady state is reached. Afterwards, another 2.5 s of physical time are simulated for statistics. The time step is either fixed at 25 ms, ensuring
a maximum Courant–Friedrich–Lewy (CFL) number of approximately 0.8, or it is adjusted automatically to ensure a CFL number below 0.95.

### 3.3 Particle dynamics simulation

For the particle dynamics simulations, an Eulerian–Lagrange approach has been chosen. This means that individual particles are tracked along their trajectories through the simulation domain described above. The trajectories are calculated according to
the following equations:

$$\frac{d\boldsymbol{x}_\mathrm{p}}{dt} = \boldsymbol{U}_\mathrm{p}, \tag{1}$$

$$m_\mathrm{p}\frac{d\boldsymbol{U}_\mathrm{p}}{dt} = \sum_i \boldsymbol{F}_i. \tag{2}$$

Here, $\boldsymbol{x}_\mathrm{p}$, $\boldsymbol{U}_\mathrm{p}$, and $m_\mathrm{p}$ are the location, velocity, and mass of a particle, $t$ is the time, and $\boldsymbol{F}_i$ are the forces acting on the
particle. The forces considered in the present work are the drag force

$$\boldsymbol{F}_\mathrm{D} = \frac{3}{4}\frac{\rho_\mathrm{f}m_\mathrm{p}}{\rho_\mathrm{p}D_\mathrm{p}}C_\mathrm{D}\left(\boldsymbol{U}_\mathrm{f}-\boldsymbol{U}_\mathrm{p}\right)|(\boldsymbol{U}_\mathrm{f}-\boldsymbol{U}_\mathrm{p})|, \tag{3}$$

and the transverse lift force due to shear lift

$$\boldsymbol{F}_\mathrm{LS} = \frac{\pi}{8}\rho_\mathrm{f}d_\mathrm{p}^3 C_\mathrm{LS}\left((\boldsymbol{U}_\mathrm{f}-\boldsymbol{U}_\mathrm{p})\times\boldsymbol{\omega}_\mathrm{f}\right), \tag{4}$$

with $\boldsymbol{U}_\mathrm{f}$, $\rho_\mathrm{f}$, and $\boldsymbol{\omega}_\mathrm{f}$ being the velocity, density, and rotation of the surrounding fluid, and $\rho_\mathrm{p}$ and $D_\mathrm{p}$ being the particle's density
and diameter. The particle Reynolds numbers are assumed to be low, thus the coefficient $C_\mathrm{D}$ is usually calculated according to Stokes (1851), and for larger Reynolds numbers according to Schiller and Naumann (1933):

$$C_\mathrm{D} = \begin{cases} \frac{24}{\mathrm{Re}_\mathrm{p}} & \text{for } \mathrm{Re}_\mathrm{p} < 0.5 \\ \frac{24}{\mathrm{Re}_\mathrm{p}}\left(1+0.15\mathrm{Re}_\mathrm{p}^{0.687}\right) & \text{for } \mathrm{Re}_\mathrm{p} \geq 0.5. \end{cases} \tag{5}$$


The particle Reynolds number is calculated as

$$\text{Re}_\text{p} = \frac{D_\text{p}|\boldsymbol{U}_\text{f} - \boldsymbol{U}_\text{p}|}{\nu_\text{f}}, \tag{6}$$

with $\nu_\text{f}$ being the kinematic viscosity of the fluid. $C_\text{LS}$ is calculated according to Mei (1992):

$$C_\text{LS} = \frac{4.1126}{\text{Re}_\text{s}^{0.5}} f(\text{Re}_\text{p}, \text{Re}_\text{s}), \tag{7}$$

with

$$f(\text{Re}_\text{p}, \text{Re}_\text{s}) = \begin{cases} \left(1 - 0.3314\beta^{0.5}\right) \exp\left(-\frac{\text{Re}_\text{p}}{10}\right) + 0.3314\beta^{0.5} & \text{for } \text{Re}_\text{p} < 40 \\ 0.0524\beta\text{Re}_\text{p} & \text{for } \text{Re}_\text{p} \geq 40, \end{cases} \tag{8}$$

and with $\beta = 0.5\frac{\text{Re}_\text{p}}{\text{Re}_\text{s}}$ and the Reynolds number of the shear flow

$$\text{Re}_\text{s} = \frac{D_\text{p}^2|\boldsymbol{\omega}_\text{f}|}{\nu_\text{f}}. \tag{9}$$

To investigate the particle growth, the particles can gain and lose mass depending on thermophysical properties at their surface:

$$\frac{dm}{dt} = 2\pi d_\text{p}\rho_{\text{v,sat}}(S - S^*)f_\text{mt}, \tag{10}$$

where $\rho_{\text{v,sat}}$ is the saturation vapor mass density, $S$ is the ambient water vapor saturation ratio, $S^*$ is the water vapor saturation

ratio at particle surface (using the Köhler equation, Wilck (1999)) and

$$f_\text{mt} = \frac{1 + \text{AKn}}{1 + \text{Kn}(\text{B}_1 + (\text{B}_2 + \text{CKn})/\alpha)} \tag{11}$$

is the mass transfer transition function. $\text{Kn} = 2\lambda_\text{g}/d_\text{p}$ is the Knudson number, with $\lambda_\text{g}$ being the mean free path length of the gas molecules. The parameter $\alpha$ is the mass accommodation coefficient and it is assumed to be 1. For the coefficients A, $\text{B}_1$, $\text{B}_2$, and C the following values are used: $\text{A} = 1$, $\text{B}_1 = 0.377$ and $\text{B}_2 = \text{C} = 4/3$ (Whitby et al., 2003). As the concentration

of particles used for experiments is currently rather small, only one-way coupling is considered, i.e., the particles' influence on the fluid phase and interactions between them are neglected.

For the particle simulations, the multi region solver as introduced above is used and extended to include particle tracking, and also the domain is the same as for the characterization simulations. After an initial 1 s of physical time, the particles are injected at the aerosol inlet for another 1 s with a concentration of according to the measurements. It is $1000\,\text{cm}^{-3}$ for the

droplet activation and growth experiments described later. This leads to about 25,000 particles which are tracked during 3.5 s of real time.

To compare the particle size distributions with their measured counterparts, the size of particles in cylinder-shaped regions in different positions are analyzed for the saved time steps. These cylinders have radii of 7 mm as this is approximately the width of the air stream that enters the welas 2300 spectrometer. The sampled regions are 25 mm long, and their axes coincides

with the z-axis.





## 4 Characterization of the flow and thermodynamic properties

The characterization efforts include measurements of the flow field and the thermodynamic parameters within the measurement section including high-resolution measurements of velocity and temperature (on the decimeter to millimeter (Kolmogorov) scale) as well as measurements of the mean relative humidity. The results will be compared to those of the Large Eddy Simula-

tions. Overall, the characterization efforts have been performed to ensure the functionality and to investigate the performance of the wind tunnel. Note that the parameter space which can be set within LACIS-T is extensive. Consequently, the characterization efforts presented here will focus on $T > 0°C$ conditions as well as on the very first meter of the measurement section (with $z_0 = 0\,\mathrm{cm}$ corresponding to the position of the aerosol inlet which is $20\,\mathrm{cm}$ downstream the turbulence grid).

### 4.1 Flow properties in the measurement section

The flow field and the turbulent flow properties have been investigated for isothermal ($T_A = T_B$) and non-isothermal ($T_A \neq T_B$) conditions with different dew point temperature settings inside the measurement section. Here, results will be presented in detail for a dry, isothermal case[1], i.e. both air flows featured the same temperature $T = 20°C$ and dew-point temperature $T_d = -15°C$. Measurements were performed at various locations underneath the aerosol inlet along the shortest distance of the measurement section by means of the hot-wire anemometer measuring the vertical velocity component at $6000\,\mathrm{Hz}$ for 5 minutes at each

location. Measurements were performed for aerosol flow turned-on and turned-off conditions in order to observe the influence of the cutting edge on the flow field. For the simulations, aerosol flow turned-off conditions have been considered only.

On the left hand side of Fig. 4, the simulated time-averaged velocity is shown for two different vertical planes and nine horizontal planes. The two vertical planes show the flow field located underneath the mid of a grid bar of the turbulence grid (left plane) and the mid of the grid openings (right plane), respectively. The nine horizontal planes are located at different

positions below the aerosol inlet which is at $z_0$. Note that the very first horizontal plane is $1\,\mathrm{cm}$ below $z_0$ while the others are located at $z_1$ to $z_8$ with $z_1 = 10\,\mathrm{cm}$ and $\Delta z = 10\,\mathrm{cm}$. Different characteristics of the flow field can be recognized. For example, the expected influence of the "cutting edge" on the flow field can be seen downstream of the aerosol inlet leading to a decrease of the flow velocity. Further, an increase of the mean velocity is visible close to the side walls, probably caused by the constriction of the cross-section leading to an acceleration of the velocity field which is strongest at the side walls. In the

following, these simulation results will be compared to measurement results.

The plots on the right hand side of Fig. 4 show – from top to bottom – the measured and simulated mean (vertical) velocity $\bar{w}$, its fluctuation in terms of the root-mean-square (rms) average $\sigma_w = \langle w'^2 \rangle^{1/2}$ (with $w' = w - \bar{w}$) and the energy dissipation rate $\varepsilon$ obtained at $z_1$. The determination of the energy dissipation rate is based on the relationship:

$$\varepsilon = \frac{1}{r}\left(\frac{S_w}{C}\right)^{3/2} \tag{12}$$

where $S_w = \langle (w'(z) - w'(z+r))^2 \rangle$ is the second-order structure function of the vertical velocity component (Wyngaard, 2010), $z$ is the vertical position, $r$ is the separation distance, $C = 2.1$ is the Kolmogorov constant and $\langle \cdot \rangle$ is the spatial average. We

---

[1]Note that we do not observe a significant difference between dry and moist conditions as well as for isothermal and non-isothermal conditions.



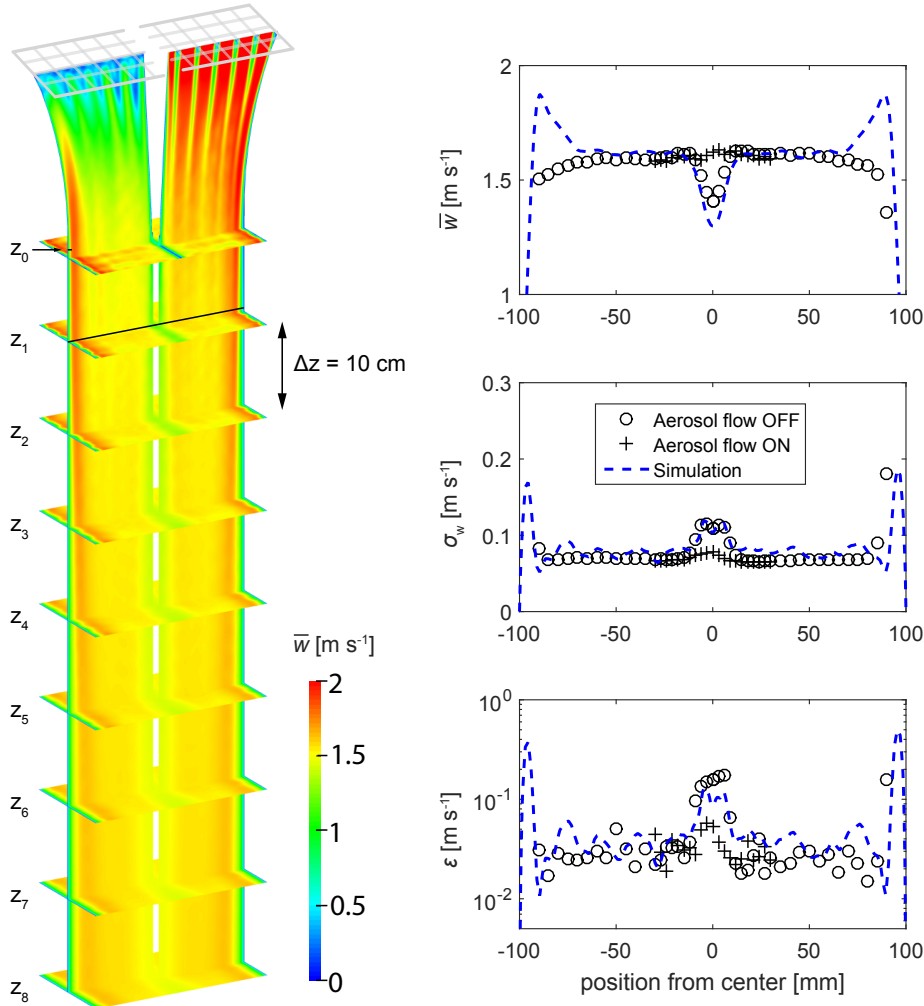

**Figure 4.** Left: Contour plot of the time-averaged velocity in two vertical and nine horizontal planes determined by the LES. The grid is also indicated. Note that the aerosol flow is turned off in the simulations. Right: Profiles of mean velocity $\bar{w}$, rms velocity $\sigma_w$ and energy dissipation rate $\varepsilon$ along the center region (black line in the left figure; position 0 mm corresponds to the center of the measurement section depth, being 100 mm away from each side wall) at $z_1 = 10$ cm. The results presented are based on hot-wire measurements (at 6000 Hz) performed in the central position of the measurement section underneath the mid of the grid openings. Measurements were taken with turned-on (crosses) and turned-off (open circles) aerosol flow. Corresponding simulation results for $\bar{w}$ and $\sigma_w$ are shown as blue dashed lines, for turned off aerosol flow only.

used Taylor's frozen flow hypothesis to transform from temporal space to physical space. The application of this hypothesis is reasonable as $\bar{w} \gg \sigma_{\mathrm{w}}$.





The mean velocity profile is homogeneous, apart from the areas near the wall of the measurement section and in the center-region with the aerosol flow being turned off. The increase of the mean velocity close to the side walls is not visible in the measurements probably due to objects related to the turbulence grid's clamping system (not considered in the simulations), which strongly reduce the acceleration at the walls. Increased turbulence intensities, represented in terms of $\sigma_w$, are observed in the near-wall area due to wall induced turbulence. Furthermore, shear stresses in the mixing region cause higher turbulence and thus higher fluctuations. By turning on the aerosol flow (isokinetic flow conditions), the inhomogeneity in the center-region of the measurement section can be eliminated for both the mean velocity and its fluctuations. Thus these shear effects are strongly decreased in the mixing zone of the two air-flows. The dissipation rate is in the range of $2.6 \times 10^{-2}\,\mathrm{m^2 s^{-3}}$ in the homogeneous region, which leads to the Kolmogorov time and length scales of $\tau_\eta = (\nu_\mathrm{f}/\varepsilon)^{1/2} = 0.02\,\mathrm{s}$ and $\eta = (\nu_\mathrm{f}^3/\varepsilon)^{1/4} = 0.6\,\mathrm{mm}$, with $\nu_\mathrm{f} = 1.5 \times 10^{-5}\,\mathrm{m/s^2}$. The integral length $l$ and the Taylor microscale $\lambda_\mathrm{t}$ are $l \sim \sigma_\mathrm{w}^3/\varepsilon = 1.3\,\mathrm{cm}$ and $\lambda_\mathrm{t} = (15\nu\sigma_\mathrm{w}^2/\varepsilon)^{1/2} = 0.6\,\mathrm{cm}$. The latter leads to the Taylor Reynolds number $Re_\lambda = \sigma_\mathrm{w}\lambda_\mathrm{t}/\nu \approx 30$. $Re_\lambda$ is much smaller than the values typically encountered in atmospheric clouds, however, this is not a limitation for studies of small-scale interactions of cloud particles/droplets and turbulence as long as intermittency aspects can be neglected (Siebert et al., 2010; Chang et al., 2016).

Figs. 5a, 5b and 5c show the values for $\bar{w}$, $\sigma_\mathrm{w}$ and $\varepsilon$ for five different distances to the aerosol inlet averaged over the range where the dissipation rate is homogeneous. Looking at the values determined through the measurements, it can be seen that the mean velocity remains almost constant (slight decrease with increasing distance to the aerosol inlet), while the fluctuations and thus also the energy dissipation rate decrease with increasing distance from the turbulence grid. This observed decrease of the turbulent kinetic energy, which is also presented in terms of $\sigma_\mathrm{w}^2$ in Fig. 5b, follows a power law function $\sigma_\mathrm{w}^2 \sim (z - z_\mathrm{grid})^n$ where $z_\mathrm{grid}$ represents the grid position. The exponent $n$ is $-1.4$ and is comparable with results reported in other wind tunnel investigations and depends on the initial conditions (Lavoie et al., 2007, and references therein). Additionally, the simulated values for $\bar{w}$, $\sigma_\mathrm{w}$ and $\varepsilon$ are given. We observe a slight decrease of $\bar{w}$ until $z = 0.2\,\mathrm{m}$ which is similar to experimental observations. However, it is followed by a slight increase which is in contrast to the measurements. This slight increase in the simulated $\bar{w}$ is caused by the high velocities obtained close to the walls (see Fig. 4), which spread slowly and also reach the inner region. The decrease of $\sigma_\mathrm{w}$ and $\varepsilon$ is well reproduced by the simulations.

The turbulent spectrum for the velocity fluctuations is shown as an example in Fig. 5d. A red line with $-5/3$ slope, which is expected for the inertial subrange of the turbulent energy cascade, is shown as a reference. The inertial subrange is not fully evolved, which is to be expected for $Re_\lambda \approx 30$. However, this is not a limitation as turbulence on the small scales is already developed (Schumacher et al., 2007) and our focus will be on small-scale interactions of cloud particles and turbulence.

## 4.2 Thermodynamic properties in the measurement section

The momentum exchange in turbulent flows is comparable to an increased molecular viscosity. However, the turbulent mixing does not only include momentum but also further associated properties such as heat and mass. We performed several characterization experiments to study the turbulent transport of heat and mass in the measurement section. All related studies were performed without the insertion of aerosol particles. The measurements are again accompanied by LES where the inlet conditions corresponded to those of the respective experiments.





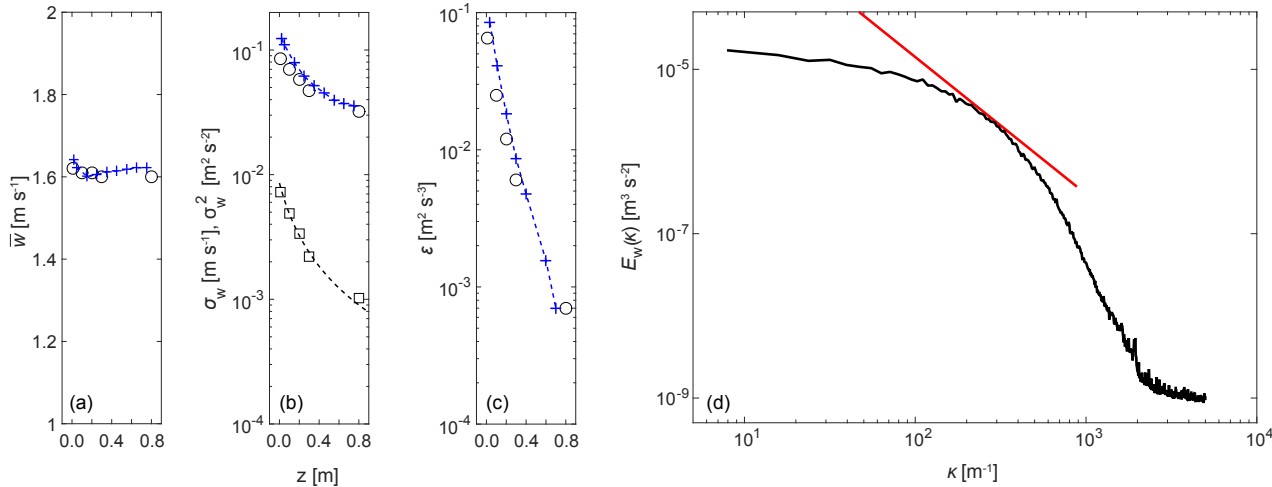

**Figure 5.** The measurements shown in Fig. 4 were carried out for five different heights below the aerosol inlet. The values for mean velocity, rms velocity and dissipation rate are averaged in the range from $-70$ mm to $+70$ mm (switched on aerosol flow) and are shown in (a), (b) and (c) given by the black circles. (b) further shows the drop in turbulent kinetic energy in terms of the squared rms velocity (black squares) which follows a power law function with an exponent of $-1.4$ (dotted black line). Additionally, the respective results from the simulations are shown (blue plus signs) being also averaged in the range from $-70$ mm to $+70$ mm. The experimentally determined turbulent spectrum for the velocity fluctuations is shown in (d). A red line with $-5/3$ slope is shown as a reference.

In the first step, the turbulent transport of heat was investigated. To do so, a temperature difference of $\Delta T = 10\,\mathrm{K}$ was set between the two flow branches ($T_A = 23°C$ and $T_B = 13°C$) and $T_d = -15°C$ in both air flows (i.e., $\Delta T_d = 0\,\mathrm{K}$).

Figure 6 shows the time-averaged temperature profiles at three different locations underneath the aerosol inlet. As expected, the turbulent mixing zone widens with increasing distance from the aerosol inlet. There is a slight increase in temperature starting at about $-70$ mm out of the center caused by heat transfer from the wall as it is in contact with ambient air on the outside which is at a temperature of $\sim 24°C$. The wall on the opposite site consequently does not significantly influence the temperature measurements because here the temperature difference between wall and flow is only $\Delta T = 1$ K. However, this wall

effect, which will be eliminated in the near future by suitable heat isolation of the measurement section, has a negligible effect on the mixing zone. In addition, the results of the simulations for the mean temperature are included in Fig. 6. The simulations reproduce the measurements in an accurate manner, albeit slightly underestimating the width of the turbulent mixing zone.

The RMS temperatures also show the increase of the width of the mixing zone with increasing $z$. Again, thermal wall effects lead to an increase of the RMS temperature towards the wall, however, having a negligible effect on the mixing zone.

After investigating the turbulent transport of heat in dry air ($T_d = -15°C$), a case with moist air is considered now. The dew point temperature was set to $12°C$ in both flow branches (i.e., $T_d$ is constant and there is no additional source or sink of water vapor), the temperature settings corresponded to those of the previous test case ($\Delta T = 10\,\mathrm{K}$ and $\Delta T_d = 0\,\mathrm{K}$) so that the relative





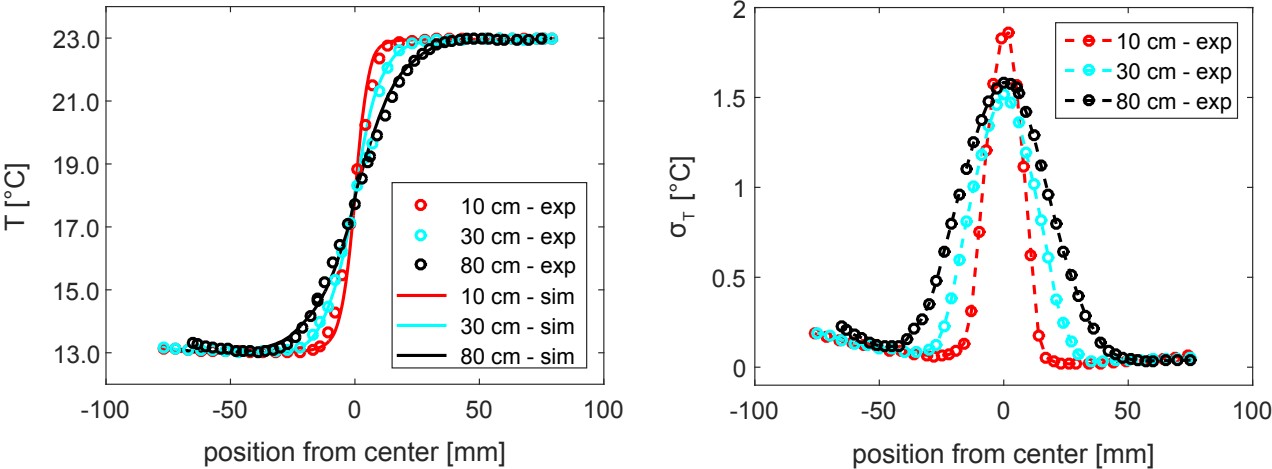

**Figure 6.** Left: Comparison of experimental temperature profiles (circles) with predictions from the simulations (lines) for $z_1$ (in red), $z_3$ (in cyan) and $z_8$ (in black). Right: Experimental results for temperature fluctuations for $z_1$ (in red), $z_3$ (in cyan) and $z_8$ (in black).

humidities correspond to $RH_A = 50\%$ and $RH_B = 93\%$. The mean dew point in the measurement section was determined with the DPM. To do so, a movable quarter-inch tubing with vertical inlet was inserted into the measurement section and

connected to the DPM. The relative humidity was calculated from the dew point temperature and the air flow temperature, based on the August–Roche-Magnus empirical formula. In the following, the time-averaged values for temperature, dew-point temperature and relative humidity are presented for the conditions at $z_3 = 30\,\mathrm{cm}$ recorded along a horizontal profile (see Fig. 7). Additionally, the results of the simulations are depicted. Since the dew point temperature along the profile is constant, i.e., the partial water vapor pressure is constant, the relative humidity essentially depends only on the temperature. The simulations

reproduce the measurements in an accurate manner.

     In the previous investigations, we focused on the turbulent transport of heat in dry and moist air. Now, the turbulent transport of mass in addition to heat is studied. To do so, a temperature and a dew point temperature difference were set between the two particle-free air flows, $\Delta T = 10\,\mathrm{K}$ and $\Delta T_\mathrm{d} = 4\,\mathrm{K}$.

     In flow branch A the dew point temperature was set to $T_{\mathrm{d,A}} = 12°\mathrm{C}$ (water vapor mixing ratio $q_{\mathrm{v,A}} = 8.68\,\mathrm{g/kg}$) and the air

flow temperature was set to $T_\mathrm{A} = 23°\mathrm{C}$ resulting in a mean relative humidity of approx. 50%. Flow branch B featured a dew point temperature of $T_{\mathrm{d,B}} = 8°\mathrm{C}$ (water vapor mixing ratio $q_{\mathrm{v,B}} = 6.64\,\mathrm{g/kg}$) and an air flow temperature of $T_\mathrm{B} = 13°\mathrm{C}$, leading to approx. 71% relative humidity. Fig. 8 shows the relative humidity profile for $z = 6\,\mathrm{cm}$, $17\,\mathrm{cm}$ and $30\,\mathrm{cm}$. The turbulent mixing zone expands with increasing distance to the aerosol inlet. The simulations, using the same inlet conditions, reproduce the measurements in an accurate manner, again slightly underestimating the width of the turbulent mixing zone as observed in

the previous cases.





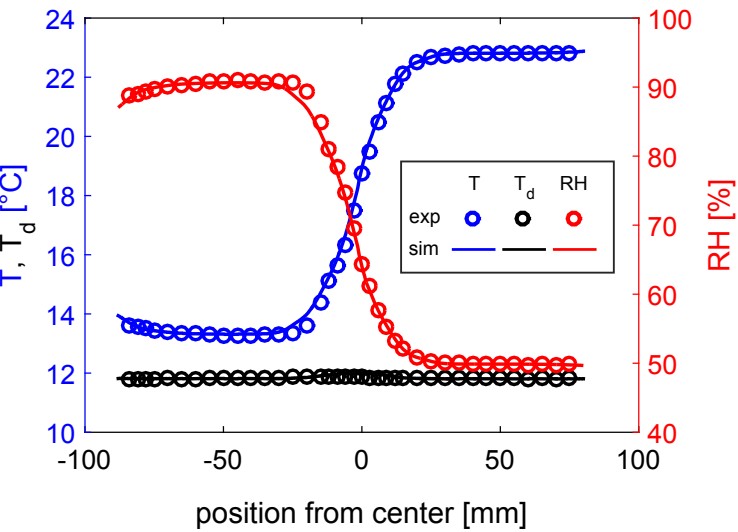

**Figure 7.** Dew point temperature (black markers), temperature (red markers), and RH profiles (blue markers) including the simulation results, for $\Delta T = 10\,\mathrm{K}$ and $\Delta T_\mathrm{d} = 0\,\mathrm{K}$ obtained at $z_3$.

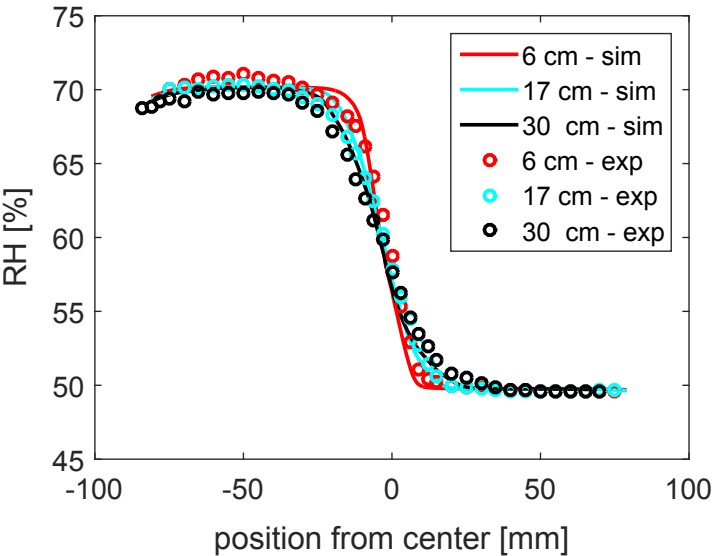

**Figure 8.** RH profiles for $\Delta T = 10\,\mathrm{K}$ and $\Delta T_\mathrm{d} = 4\,\mathrm{K}$ based on temperature and dew point temperature measurements for $z = 6\,\mathrm{cm}$, $17\,\mathrm{cm}$, and $30\,\mathrm{cm}$ including numerical predictions.





Finally, on the left hand side of Fig. 9 we compare the temperature profile ($\Delta T = 10\,\text{K}$) for the different conditions, i.e., dry with $\Delta T_\text{d} = 0\,\text{K}$; moist with $\Delta T_\text{d} = 0\,\text{K}$; and moist with $\Delta T_\text{d} = 4\,\text{K}$, exemplarily for $z_3$. As expected, the temperature profiles shown in the left of Fig. 9 exhibit a very similar behavior, i.e., the influence of the increased amount of water vapor in the air flow as well as the water vapor profile itself (in terms of RH) on the temperature curve is very low. The reasons are that still

less than 1% of the total mass is water vapor which does not significantly influence the fluid properties (e.g., heat capacity). Further, there is no condensation of water vapor which could influence the temperature profile due to latent heat release.

On the right hand side of Fig. 9 the temperature and water vapor mixing are shown for the moist case ($\Delta T_\text{d} = 4\,\text{K}$). In order to compare both quantities, the normalized water vapor mixing ratio $\xi_\text{n}$ and the normalized temperature $\theta_\text{n}$ are depicted. The normalized water vapor mixing ratio is defined as $\xi_\text{n} = (q_\text{v} - q_{\text{v},1})/(q_{\text{v},2} - q_{\text{v},1})$, where $q_{\text{v},1}$ and $q_{\text{v},2}$ are lowest and

highest water vapor mixing ratio set in the respective flow branch, respectively. The normalized temperature is given through $\theta_n = (T - T_1)/(T_2 - T_1)$ with $T_1$ being the lowest and $T_2$ being the highest temperature set in the respective flow branch. Both curves fall together, i.e, $\xi_\text{n}$ and $\theta_\text{n}$ behave similarly as we would expect since turbulent transport processes dominate over laminar diffusion processes in the mixing zone.

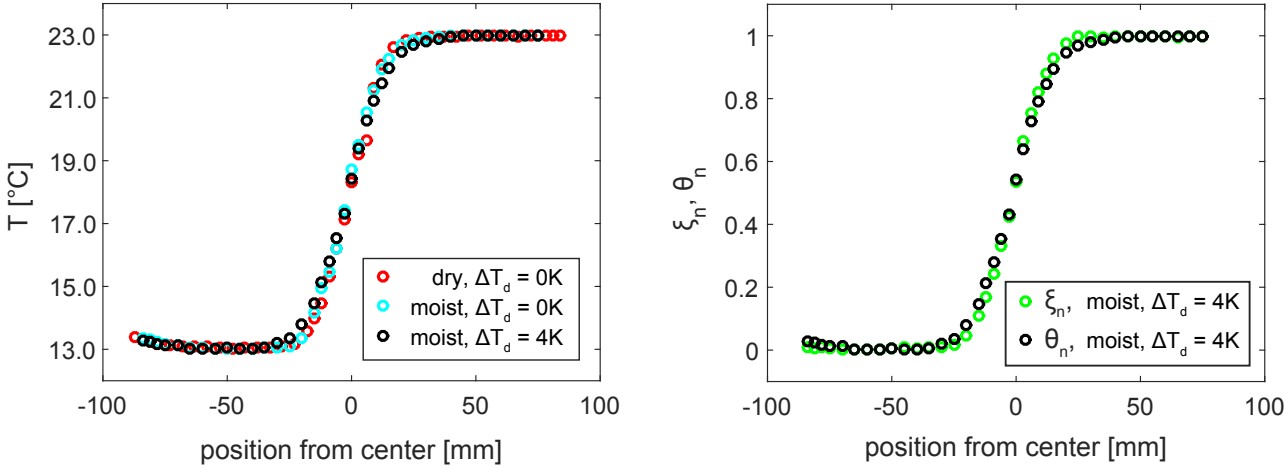

**Figure 9.** Left: Mean temperature profiles for three different cases at $z_3$. Right: Normalized mean temperature and mean mass fraction at $z_3$ for $\Delta T = 10\,\text{K}$ and $\Delta T_\text{d} = 4\,\text{K}$.

In summary, the above described investigations and results clearly demonstrate the functionality of LACIS-T. The current

setup creates sufficiently large regions of homogeneous velocities in the measurement section. Determined dissipation rates are similar to those in atmospheric clouds. The decrease of the turbulent kinetic energy with increasing distance from the turbulence grid is comparable with other wind tunnels. Further, the turbulent mixing behavior of heat in dry air and moist air, as well as the turbulent mixing behavior of water vapor indicate that the transport of heat and mass in the mixing zone





are governed by turbulent processes whereas laminar processes become negligible. Altogether, we observe a well-defined and

controllable turbulent mixing process that can be simulated accurately.

## 5 First experimental results on particle deliquescence/hygroscopic growth and droplet activation/growth

The first experiments conducted at LACIS-T deal with the deliquescence, hygroscopic growth and activation of size-selected, monodisperse aerosol particles under turbulent conditions. Sodium chloride particles (NaCl) were used for both experiments. Two different settings have been applied for the studies which will be described accordingly.

### 380 5.1 Deliquescence and Hygroscopic growth

In the first experiment, the deliquescence and hygroscopic growth behavior of NaCl particles with a dry diameter $D_{\mathrm{p,dry}}$ of 320 nm was investigated. Both particle-free air flows featured the same conditions in terms of flow rate, temperature and dew-point temperature. The temperature and dew-point temperature of the aerosol flow were independently adjusted compared to the two particle-free air flows. The aerosol flow rate was set to enter the measurement section in isokinetic fashion (the inlet

concentration was 1000 particles/cm$^3$). The welas 2300 sensor for particle detection was positioned inside the measurement section at $z_3 = 30$ cm right below the aerosol inlet. The temperature of both particle-free air flows was set to 20°C. The dew point temperature was varied between 14.4°C and 19.8°C resulting in relative humidities (RH) in the measurement section between 68% and 98%. The particles were introduced into the wind tunnel either dry ($T_{\mathrm{d}}$ = -15°C and $T$ = 20°C) or already pre-moistened ($T_{\mathrm{d}}$ = 19°C and $T$ = 20°C), i.e., we investigated the hygroscopic growth of non-deliquesced and deliquesced

NaCl particles, respectively.

Figure 10 shows the measured particle diameter versus the relative humidity (obtained through measurements of dew-point and air temperature). The blue solid line represents the corresponding Köhler curve (Pruppacher and Klett, 1997). We observed deliquescence of the NaCl particles at approx. 75% RH, which compares well to literature data (e.g., 75.7% measured at 25°C by Tang et al. (1977)). Furthermore, deliquescence is observed over a range of RH, which is indicative for an influence of the

prevailing turbulent RH fluctuations. Note that the investigations were performed at a total flow rate of 10,000 l/min, which makes the accuracy of the results even more impressive.

### 5.2 Droplet activation and growth

In the second experiment, droplet formation on size-selected, monodisperse NaCl particles with $D_{\mathrm{p,dry}}$ of 100 nm, 200 nm, 300 nm, and 400 nm and the subsequent droplet growth were investigated. To do so, a temperature difference of $\Delta T = 16$ K

was set between the two particle-free air flows. The temperature and dew-point temperature of the air streams were set to 20°C in branch A and 4°C in branch B, respectively, so that RH = 100% in each air flow. Due to the mixing of both saturated air flows in the measurement section, supersaturation conditions are reached. Based on the simulations (not shown), the mean RH was at approximately 101.5%.





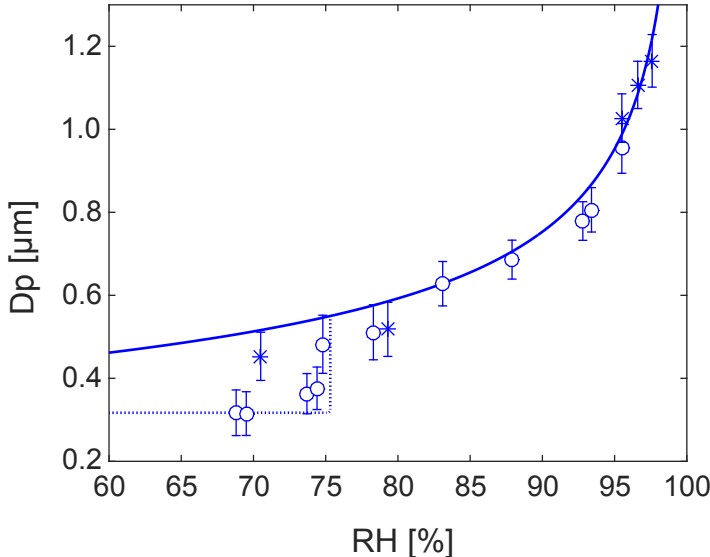

**Figure 10.** Hygroscopic growth of deliquesced (asterisk) and undeliquesced (open circles) NaCl particles with $D_{p,dry} = 320\,nm$ at $T = 20°C$ with welas 2300. As reference the theoretical Köhler curve for deliquesced particles (solid line) as well as the deliquescence point (dashed line, deliquescence at 75.7% measured at 25°C by Tang et al. (1977)) are shown. We observe deliquescence of NaCl particles at about 75%.

For each injected $D_{p,dry}$, the particle concentration was set to $1000\,cm^{-3}$. The welas 2300 sensor was positioned at center

position inside the measurement section at $z_4 = 40\,cm$ or $z_8 = 80\,cm$, in order to determine the prevailing droplet size distributions. The determined size distributions at the two positions are shown in Fig. 11. In both figures, the normalized droplet number vs. the particle diameter is displayed. The following observations can be made: a) For each $D_{p,dry}$ the formed droplets grow with increasing distance to the aerosol inlet; b) all the size distributions nearly fall together at $z_8$ (see Fig. 11b); c) the size distributions are negatively skewed and d) we also observe a significant number of particles close to $D_p = 300\,nm$, which

is approximately the welas 2300 detection limit.

To start with the interpretation of these observations, we included the critical diameters $D_{p,crit}$ for particle activation which are $1.2\,\mu m$, $3.4\,\mu m$, $6.3\,\mu m$, and $9.7\,\mu m$ (dotted lines in Fig. 11) for $D_{p,dry} = 100\,nm$, $200\,nm$, $300\,nm$, $400\,nm$, respectively. For all dry particle sizes investigated, the supersaturation reached inside the measurement section is high enough to activate these particles to cloud droplets. However, only the grown droplets which originate from the $D_{p,dry} = 100\,nm$ and $D_{p,dry} = 200\,nm$

particles are almost all or mostly activated at $z_8$ while the ones formed on the $D_{p,dry} = 300\,nm$ and $D_{p,dry} = 400\,nm$ particles are mostly not. The reason for this observation is the kinetic limitation of droplet growth. The time the particles are exposed to a certain level of supersaturation must be long enough to reach the respective critical diameter (Chuang et al., 1997; Nenes et al., 2001). For the $D_{p,dry} = 300\,nm$ and $400\,nm$ particles and the prevailing supersaturation, it is on the order of several ten's of seconds. However, the time to reach $z_8$ is about 0.5 s which is too short for these particles to reach their respective $D_{p,crit}$.





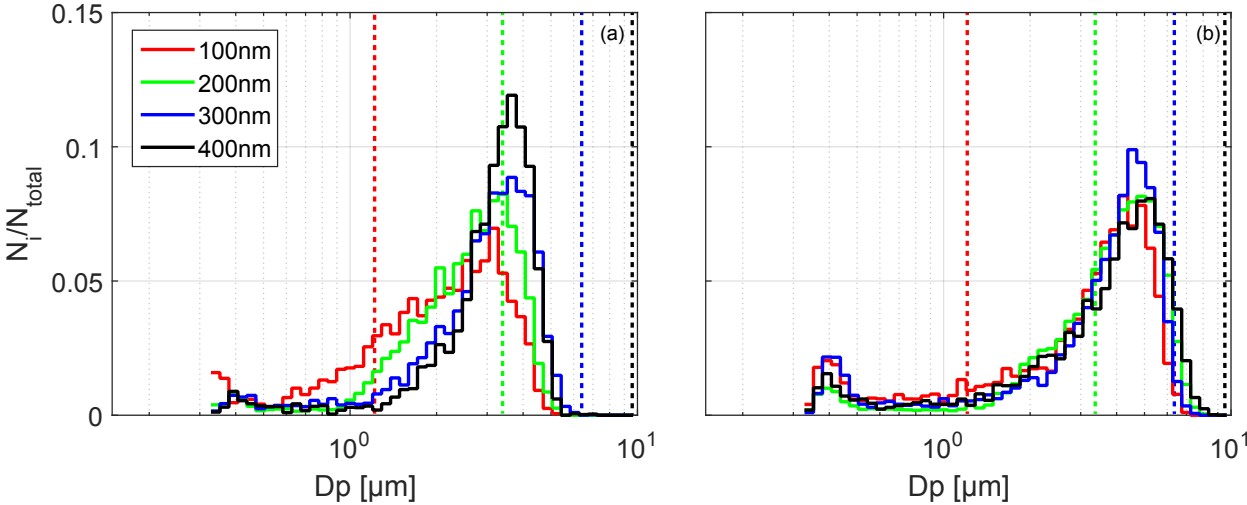

**Figure 11.** Droplet formation and growth of differently size-selected, monodisperse NaCl particles ($D_{\mathrm{p,dry}}$ = 100 nm - 400nm) for $\Delta T$ = 16 K measured at two different positions below the aerosol inlet (left figure: $z_4$ = 40 cm, right figure: $z_8$ = 80 cm). The dotted lines represent the critical diameters $D_{\mathrm{p,crit}}$ for particle activation which are 1.2 $\mu$m, 3.4 $\mu$m, 6.3 $\mu$m, and 9.7 $\mu$m for $D_{\mathrm{p,dry}}$ = 100 nm, 200 nm, 300 nm, 400 nm, respectively.

Naturally, it also limits the further growth of the droplets which formed on the $D_{\mathrm{p,dry}}$ = 100 nm and $D_{\mathrm{p,dry}}$ = 200 nm particles as for the diffusional growth it is irrelevant whether the droplets are activated or not as long as the supersaturation is above the critical value which depends on the dry particle size. In conclusion, under the prevailing conditions and the sole observation of the grown droplet distributions, it is not possible to distinguish between the activated and non-activated droplet distributions or to determine which distribution represents the activated and which the not-activated state. In other words, droplet growth is

kinetically limited regardless if the droplets are in the hygroscopic or dynamic growth regime. Further, the dry particle size is of minor importance for the observed droplet distributions, especially with increasing residence time.

    In order to interpret the negative skewness of the distributions as well as the significant number of particles close to $D_{\mathrm{p}}$ = 300 nm, we consider the LES results which are shown in Fig. 12. In the left figure, a snapshot of the instantaneous saturation field in the symmetry plane as well as the respective particle diameters grown on $D_{\mathrm{p,dry}}$ = 100 nm NaCl particles

along the vertical axis are shown. For $z_4$ and $z_8$, droplet size distributions are extracted from the simulations and displayed together with the measured size distributions in the right plots of Fig. 12. From the simulations, the magnitude of the RH fluctuations in terms of a standard deviation can be determined to be $\sigma_{\mathrm{RH}}$ = $\sim 4\%$.

    First of all, the simulations reproduce the measurements in an accurate manner. At $z_4$, the simulated droplet distribution is of bimodal shape where the left shoulder of the first mode was not detected by the welas 2300 due to its detection limit. Further the

negative skewness can be observed in both sub-figures. From the simulation of individual particle tracks (not shown) it can be concluded that the small particles ($D_{\mathrm{p}} < D_{\mathrm{p,crit}}$) are hygroscopically grown particles that did not experience supersaturated





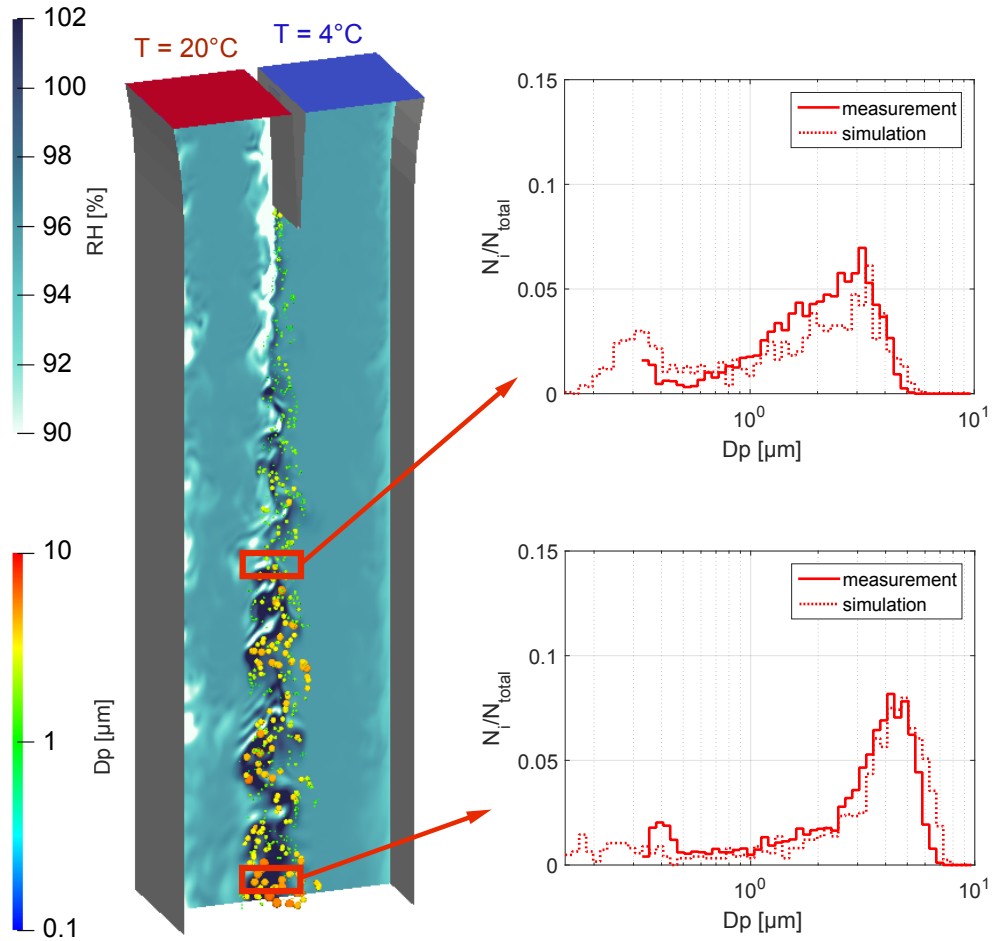

**Figure 12.** Snapshot of particle simulation with fluctuating relative humidity field in the background and particles colored and sized (not to scale) according to their diameter. Right: Droplet formation and growth of monodisperse NaCl particles ($D_{\mathrm{p,dry}} = 100\,\mathrm{nm}$) for $\Delta T = 16\,\mathrm{K}$ measured at two different positions below the aerosol inlet (upper right figure: $z_4$, lower right figure: $z_8$) for measurements (solid line) and simulations (dashed line).

conditions but also droplets that deactivated because they experienced sub-saturated conditions in the fluctuating saturation field. In general, these turbulent fluctuations in RH broaden the droplet size distribution towards smaller diameters due to evaporating droplets or less-grown droplets in left tail of the droplet size distribution. In other words, the negative skewness of

the obtained droplet size distributions is indicative for turbulence-influenced droplet formation and growth / evaporation. The particle size plays a minor role here.



## 6 Summary and outlook

We have developed the turbulent moist-air wind tunnel LACIS-T, specifically aiming at a better understanding of aerosol–cloud–turbulence interactions. The advantage of LACIS-T in particular and laboratory experiments in general is that specific, atmospherically relevant processes can be studied under well controlled and repeatable initial and boundary conditions, whereas in field experiments, there often is significant uncertainty in the measurements themselves, in the boundary conditions, and in the statistical stationarity of conditions (Chang et al., 2016). Furthermore, laboratory experiments can provide scenarios for which physical theory and models can be directly compared to an experiment, with known initial and boundary conditions (Stratmann et al., 2009). Therefore, laboratory experiments, in which the turbulence and thermodynamic conditions are reliably reproducible, and long-term averaging of measurements under statistically stationary conditions can be achieved, are invaluable for increasing our quantitative understanding concerning atmospheric cloud processes.

The investigations described here show that LACIS-T is suitable for studying the influence of turbulent temperature and water vapor fluctuations on cloud microphysical processes. We observed deliquescence to take place over a range of mean RH, which is indicative of a prevailing influence of turbulent RH fluctuations, also noting the precision at high volume flow rates. We further obtained indications of the influence of turbulent supersaturation fluctuations on the droplet activation. Concerning the latter, our results also suggest that kinetical effects and/or limitations may be important in inhibiting droplet activation in a turbulent environment. On the other hand, turbulence can also lead to the occurrence of locally high supersaturations, which together with the highly non-linear Köhler- and condensational growth equations, might increase the number of activated droplets. Altogether, turbulent fluctuations affect the droplet size distribution. Our first results are very promising in terms of the ability to capture the observed processes with the LES, as well as the ability to see clear indications of the effect of turbulence on droplet activation and growth. These results will be verified and quantified in more detail in the near future. In that sense we will also focus on the relative roles of turbulence vs. aerosol particle physical and chemical properties (particle size, number and composition).

We further aim at gaining fundamental and quantitative understanding of the influences of entrainment and detrainment processes on the microphysical properties of clouds as well as the influence of turbulence on heterogeneous ice nucleation in the near future. LACIS-T is also part of the EU-funded (HORIZON 2020) infrastructure project EUROCHAMP-2020 which is planned to be embedded into ACTRIS (Aerosol, Cloud and Trace Gases Research Infrastructure). In the scope of the infrastructure projects, LACIS-T is being made available to other scientists from atmospheric science community to address interdisciplinary problems as well as to use the wind tunnel for scientific instrument testing and calibration.

In summary, results from LACIS-T investigations will have the potential to help interpreting and corroborating the results from related in-situ measurements in clouds (e.g., Ditas et al., 2012), and therefore enhance our understanding of the interactions between cloud microphysics and turbulence, and consequently cloud processes in general.



*Data availability.* The experimental data is available via the EUROCHAMP-2020 data center as well as upon request to the contact author. For numerical data please contact S. Schmalfuß (schmalfuss@tropos.de) or D. Niedermeier.

*Author contributions.* D. Niedermeier and S. Schmalfuß (section 3) wrote the manuscript with contributions from all co-authors. LACIS-T measurements and data evaluation were performed by D. Niedermeier, D. Busch, J. Voigtländer, and S. Schmalfuß. Numerical simulations were performed by S. Schmalfuß with contributions from D. Niedermeier, J. Voigtländer, and F. Stratmann. All authors discussed the experimental and numerical results. F. Stratmann initiated and conducted the conceptualization, planning and buillt-up of LACIS-T with significant contributions from J. Voigtländer, J. Schumacher, and R. A. Shaw.

*Competing interests.* The authors declare that they have no conflict of interests.

*Acknowledgements.* LACIS-T was constructed in the framework of the Leibniz-SAW-Project "Leipzig Aerosol Cloud Turbulence Tunnel" (number: SAW-2013-IfT-2). This project/work has also received funding from the European Union's Horizon 2020 research and innovation program through the EUROCHAMP-2020 Infrastructure Activity under grant agreement No 730997. D. Niedermeier acknowledges support from the Alexander von Humboldt Foundation.



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
