# Peer review of "Characterization and first results from LACIS-T: A moist-air wind tunnel to study aerosol-cloud-turbulence interactions"

_Atmospheric Measurement Techniques, 2019_

## Referee Comment (RC1) · Anonymous Referee #2 · 3 Dec 2019

General Comment: This manuscript presents the newly developed turbulent moist-air wind tunnel, called the Turbulent Leipzig Aerosol Cloud Interaction Simulator (LACIS-T). LACIS-T is able to study different cloud processes taking into account interactions between turbulence and cloud microphysical processes. Additionally, the authors complemented their LACIS-T experiments with Computational Fluid Dynamics (CFD) simulations to explain their observations. The behavior of the LACIS-T was tested by performing deliquescence and hygroscopic growth as well as droplet activation and growth experiments using NaCl particles.

This is as well written manuscript, with a very detailed descriptions of this newly devel-

oped turbulent moist-air wind tunnel. The LACIS-T is a great and valuable instrument for the cloud physics community that can be used to fulfill many gaps in knowledge. Given the lack of instruments like this, LACIS-T can have a huge impact in the near future. I congratulate the authors for developing such a great instrument and for the careful characterization. I only have one "Major Comment". The manuscript can be accepted after the following minor comments are added to the revised manuscript.

Major Comment: It would have been nice to add a reference experiment, especially for the droplet activation experiments. I mean, is it possible to run a droplet activation experiment under steady conditions, i.e., without any turbulence? This will show how monodisperse is the droplet size distribution (DSD) in comparison to the DSD shown in Figure 12.

Minor Comments: L19: Add a reference after "Earth". L20: Add a reference after "interactions". L24: Add a reference after "scales". L28: I suggest to add other references in addition to Siebert et al. (2006). L28: "It links to phase transition processes". Do the authors refer to "turbulence"? L34: Add a reference after "undertaking". L37: I suggest to add other references in addition to Stratmann et al. (2009). L40: How about Cziczo et al. (2017)? L44-49: I do not think it is necessary to cite all this previous papers. L50: I think "those of the other" should be "those of other". L51: Add a reference after "interactions". L62-73: Much of the information provided here can go into methods. L104: "to remove aerosol particles". In the particle-free air? L137: "Condensational" should be "Condensation". L140: Delete "and" before 200. L259-260: "Large Eddy Simulations" should be "LES". L302: I suggest to change it to "Figs. 5a-c" L333 and 335: "RMS" should be in lowercase? L398: "size-selcted" should be "size-selected".

Reference: Cziczo, D. J.; Ladino, L. A.; Boose, Y.; Kanji, Z. A.; Kupiszewski, P.; Lance, S.; Mertes, S.; Wex, H. Measurements of Ice Nucleating Particles and Ice Residuals. In Ice Formation and Evolution in Clouds and Precipitation: Measurement and Modeling Challenges; American Meteorological Society, 2017; Vol. 58, pp 8.1−8.13

---

## Referee Comment (RC2) · Anonymous Referee #1 · 6 Jan 2020

The design and performance of a new system for studying turbulent effect on cloud microphysics under short timescales up to a few seconds is presented, the LACIS-T system. CFDC simulations using large eddy simulations are also performed to help interpret the system's performance and experimental results. This is an important contribution to the atmospheric science community given the lack of experimental systems to directly study the effects of turbulence. I recommend it for publication, and make a few suggestions to further improve the manuscript below.

While the system's design and characterization are nicely presented in great detail, it stuck me that a discussion of how finely the revenant parameters can bee adjusted and

[Figure]

controlled in LACIS-T was not really presented. This would be a valuable addition to the paper.

In the introduction, more elaboration on the importance of atmospheric turbulence and its effects on important properties and phenomena such as cloud microphysics and particle deliquescence/growth and the resulting rather complex and intriguing size distributions is warranted. This will be the results presented in better context.

Line 91: Is there a reference for the "Göttingen type" of wind tunnel?

Lines 205-210: To make this more accessible to those less familiar with CFD simulations, please explain "periodic boundary conditions" and the significance and utility of the Courant–Friedrich–Lewy (CFL) number.

Line 225: What is the relevant particle size range and fluid velocity range being considered when evaluating the particle Reynolds number?

Fig. 1: Please indicate the air flow direction in the various channels.
* * *

---

## Author Comment (AC1) · 9 Mar 2020

**Referee 2**

*General Comment: This manuscript presents the newly developed turbulent moist-air wind tunnel, called the Turbulent Leipzig Aerosol Cloud Interaction Simulator (LACIST). LACIS-T is able to study different cloud processes taking into account interactions between turbulence and cloud microphysical processes. Additionally, the authors complemented their LACIS-T experiments with Computational Fluid Dynamics (CFD) simulations to explain their observations. The behavior of the LACIS-T was tested by performing deliquescence and hygroscopic growth as well as droplet activation and growth experiments using NaCl particles. This is as well written manuscript, with a very detailed descriptions of this newly developed turbulent moist-air wind tunnel. The LACIS-T is a great and valuable instrument for the cloud physics community that can be used to fulfill many gaps in knowledge. Given the lack of instruments like this, LACIS-T can have a huge impact in the near future. I congratulate the authors for developing such a great instrument and for the careful characterization. I only have one "Major Comment". The manuscript can be accepted after the following minor comments are added to the revised manuscript.*

We thank referee 2 for his/her remarks/comments/suggestions. They are addressed below and we have revised the manuscript accordingly.

*Major Comment: It would have been nice to add a reference experiment, especially for the droplet activation experiments. I mean, is it possible to run a droplet activation experiment under steady conditions, i.e., without any turbulence? This will show how monodisperse is the droplet size distribution (DSD) in comparison to the DSD shown in Figure 12.*

We agree that such a reference experiment would be very valuable. Therefore, **as kind of a benchmark, we want to evaluate how the droplet size distribution would look like without turbulence affecting droplet formation and growth. Therefore, utilizing the above described numerical model, two additional cases have been investigated, a) a case without grid induced turbulence (i.e., the grid was removed from the numerical simulation) and b) an idealized case based on time-averaged flow fields without turbulent fluctuations. In these simulations, the formation and growth of NaCl particles, with $D_{p,dry}$ = 100 nm for the temperature difference of $\Delta T$ = 16K was considered. The simulation results are shown in Fig. 13 for $z_8$ = 80cm together with measurement results, for which the turbulence grid remains included as shown in Fig. 12.**

[Figure]

**Figure 13.** *Comparison between different model calculations for the particle formation and growth on NaCl particles with $D_{p,dry}$ = 100 nm at $z_8$ = 80cm for $\Delta T$ = 16K. Left figure: LES with turbulence grid (as shown in lower right plot of Fig. 12). Middle figure: LES but without turbulence grid. Right figure: simulation with averaged fields used as frozen flow fields, and transient particle calculation. In all plots, the measurement results, for which the turbulence grid is included (as shown in Fig.12), are shown for reference.*

**It turns out that the removal of the turbulence grid does not lead to laminar conditions. We still observe inherent turbulent conditions due to wall effects and the high Reynolds number (order of $10^4$) for the set velocity. But the turbulence intensity and therefore the strength of turbulent fluctuations is decreased. The power spectra obtained for the configuration without grid further suggest that the turbulence is anisotropic in this case (not shown). As a consequence, we still obtain a broad droplet size distribution (see middle plot in Fig. 13) which is however narrower compared to the measurement / simulation with turbulence grid. We further observe a significant number of particles close to $D_p$ = 300 nm.**
**Laminar conditions, which would lead to a very narrow droplet size distribution (see right plot in Fig. 13), can only be simulated if averaged fields are used as frozen flow fields in the simulations, which is not realizable in the real experiment. In other words, measurements without turbulence are not executable inside LACIS-T and the flow regime is best controlled in presence of the turbulence grid. Furthermore, these simulations clearly indicate the distinct influences of turbulence on the droplet size distributions, and consequently, on the formation and growth of droplets inside LACIS-T.**
The text marked in bold as well as Fig. 13 have been added at the end of section 5.2.

*Minor Comments:*
*L19: Add a reference after "Earth".*

Lamb and Verlinde (2011) has been added to the text.

*L20: Add a reference after "interactions".*

Mason and Ludlam (1951), Hobbs (1991), and Kreidenweis et al. (2019) have been added to the text.

*L24: Add a reference after "scales".*

The sentence has been connected with the followed-up sentence in order to avoid citing the same reference twice.

Atmospheric clouds are often non–stationary, inhomogeneous, intermittent, and cover an enormous range of spatial (micrometers to hundreds of kilometers) and temporal (microseconds to hours and days) scales.  with cross–scale interactions between turbulent fluid dynamics and cloud microphysical processes  influencing cloud behavior and cloud development (Bodenschatz et al., 2010).

*L28: I suggest to add other references in addition to Siebert et al. (2006).*

The sentence has been changed slightly and additional references have been added: Turbulence drives processes such as entrainment and mixing, leading to strong fluctuations in aerosol particle concentration, temperature, water vapor, and consequently supersaturation  **with implications for** cloud droplet activation, growth and decay (Siebert et al., 2006; Chandrakar et al., 2016, Siebert et al., 2017).

*L28: "It links to phase transition processes". Do the authors refer to "turbulence"?*

Yes, this is right. However, due to a comment by reviewer 1, the sentence has been

rewritten: "[…] **Turbulence also influences particle collision rates and is therefore thought to be central to precipitation formation (Shaw, 2003; Wang and Grabowski, 2009)**  […]".

*L34: Add a reference after "undertaking".*

Stratmann et al. (2009) has been added.

*L37: I suggest to add other references in addition to Stratmann et al. (2009).*

List et al. (1986) and Kreidenweis et al. (2019) have been added.

*L40: How about Cziczo et al. (2017)?*

The citation Cziczo et al. (2017) has been added to the text.

*L44-49: I do not think it is necessary to cite all this previous papers.*

We agree. Now in almost all cases, two papers are cited per particle type/species.

*L50: I think "those of the other" should be "those of other".*

We agree, it has been changed accordingly.

*L51: Add a reference after "interactions".*

Chang et al. (2016) has been added to the text.

*L62-73: Much of the information provided here can go into methods.*

This part has been written in order to get a first impression about the set-up and benefits of LACIS-T as well as answering the question what distinguishes this wind tunnel from other facilities like the PI chamber. Therefore, we would like to leave the main part here only deleting the last sentence of this paragraph.

*L104: "to remove aerosol particles". In the particle-free air?*

The sentences have been re-written:
Two radial blowers (NICOTRA-Gebhardt, Germany) separately drive the two  dry air flows (flow branches 'A' and 'B'). Flow rates of up to 6.000 l/min in each flow branch are possible. Afterwards, each flow passes a particle filter (Filter class U16; TROX GmbH, Germany) to remove aerosol particles. Subsequently, a defined amount of water vapor can be added to each of the **now** particle-free air flows by means of a humidification system.

*L137: "Condensational" should be "Condensation".*

Done.

*L140: Delete "and" before 200.*

Done.

*L259-260: "Large Eddy Simulations" should be "LES".*

Done.

*L302: I suggest to change it to "Figs. 5a-c"*

Done.

*L333 and 335: "RMS" should be in lowercase?*

Done.

*L398: "size-selcted" should be "size-selected".*

Done.

*References*

Chandrakar, K. K., Cantrell, W., Chang, K., Ciochetto, D., Niedermeier, D., Ovchinnikov, M., Shaw, R. A., and Yang, F.: Aerosol indirect effect from turbulence-induced broadening of cloud-droplet size distributions, Proceedings of the National Academy of Sciences, 113, 14 243–14 248, https://doi.org/10.1073/pnas.1612686 113, 2016.

Cziczo, D. J., Ladino, L., Boose, Y., Kanji, Z. A., Kupiszewski, P., Lance, S., Mertes, S., and Wex, H.: Measurements of ice nucleating particles and ice residuals, Meteorological Monographs, 58, 8.1–8.13, https://doi.org/10.1175/AMSMONOGRAPHS–D–16–0008.1, 2017.

Hobbs, P. V.: Research on clouds and precipitation: Past, present and future, Part II, Bulletin of the American Meteorological Society, 72, 184–191, https://doi.org/10.1175/1520–0477(1991)072,0184:ROCAPP.2.0.CO;2, 1991.

Lamb, D. and Verlinde, J.: Physics and chemistry of clouds, Cambridge University Press, Cambridge, UK, 2011.

List, R., Hallett, J., Warner, J., and Reinking, R.: The Future of Laboratory Research and Facilities for Cloud Physics and Cloud Chemistry: Report on a Technical Workshop Held in Boulder, Colorado, 20–22 March 1985, Bulletin of the American Meteorological Society, 67, 1389–1397, https://doi.org/10.1175/1520–0477–67.11.1389, 1986.

Mason, B. J. and Ludlam, F. H.: The microphysics of clouds, Reports on progress in physics, 14, 147–195, https://doi.org/10.1088/0034–4885/14/1/306, 1951.

Siebert, H. and Shaw, R. A.: Supersaturation fluctuations during the early stage of cumulus formation, Journal of the Atmospheric Sciences, 74, 975–988, https://doi.org/10.1175/JAS–D–16–0115.1, 2017.

---

## Author Comment (AC2) · 9 Mar 2020

**Referee 1**

*The design and performance of a new system for studying turbulent effect on cloud microphysics under short timescales up to a few seconds is presented, the LACIS-T system. CFDC simulations using large eddy simulations are also performed to help interpret the system's performance and experimental results. This is an important contribution to the atmospheric science community given the lack of experimental systems to directly study the effects of turbulence. I recommend it for publication, and make a few suggestions to further improve the manuscript below.*

We thank referee 1 for his/her remarks/comments/suggestions. They are addressed below and we have revised the manuscript accordingly.

*While the system's design and characterization are nicely presented in great detail, it stuck me that a discussion of how finely the revenant parameters can be adjusted and controlled in LACIS-T was not really presented. This would be a valuable addition to the paper.*

The accuracy for flow rate, temperature and dew-point temperature adjustments have been added in section 2 where the corresponding devices are described (changes are given in bold here). For example, **the accuracy of the temperature adjustments** in the humidification system and heat exchangers as well as of their monitoring **is +/- (0.03°C +0.0005 x *T*) with *T* being the actual temperature (in °C)**. Note that, **the accuracy for monitoring the set dew-point with the dew-point mirror is ≤ ±0.1 K with a reproducibility of ≤ ±0.05 K**. The **Huber thermostats** used in the humidification system and connected to the heat exchangers **feature a temperature stability of +/- 0.01 K at -10°C**. The volume flow rate is monitored by means of **ultrasonic flow meters each of which features an accuracy of 1.5% of the reading**. **The relative measurement uncertainty of the Hot-wire anemometer is about 3%**.

*In the introduction, more elaboration on the importance of atmospheric turbulence and its effects on important properties and phenomena such as cloud microphysics and particle deliquescence/growth and the resulting rather complex and intriguing size distributions is warranted. This will be the results presented in better context.*

A respected paragraph has been inserted which is marked in bold:
"Turbulence drives processes such as entrainment and mixing, leading to strong fluctuations in aerosol particle concentration, temperature, water vapor, and consequently supersaturation which affects having implications for cloud droplet activation, growth and decay (Siebert et al., 2006; Siebert and Shaw, 2017; Chandrakar et al., 2016). **Indeed, it has been shown that representation of unresolved fluxes in large-eddy simulations influence properties of simulated stratocumulus clouds (Shi et al., 2018), and that the range of scales captured in direct numerical simulations of cloud entrainment influence the width of the droplet size distribution (Kumar et al., 2018). Even without the presence of strong entrainment, fluctuations in supersaturation can influence the functional form of the cloud droplet size distribution (e.g., McGraw and Liu, 2006; Chandrakar et al., 2016; Saito et al., 2019; Chandrakar et al., 2019). Turbulence also influences particle collision rates and is therefore thought to be central to precipitation formation (Shaw, 2003; Wang and Grabowski, 2009).** These processes, in turn, can have buoyancy and drag effects on turbulence and influence cloud dynamic processes up to the largest scales (Stevens et al, 2005; Malinowski et al., 2008; Bodenschatz et al., 2010)".

*Line 91: Is there a reference for the "Göttingen type" of wind tunnel?*

The Göttingen wind tunnel goes back to Prandtl and Betz. It is a closed-loop wind-tunnel with a measurement section. A citation was added to the text: Randers-Pehrson (1935) which gives a review about wind tunnels of that time. There it is written: "With the construction of the first wind tunnel at Göttingen we are approaching modern times. This was the first return-flow tunnel, built by Dr. Ludwig Prandtl for Motorluftschiffstudiengesellschaft and completed in July 1908. This tunnel was superseded in 1916-17 by a much larger tunnel with open jet and return flow, which is now called the Göttingen type."

*Lines 205-210: To make this more accessible to those less familiar with CFD simulations, please explain "periodic boundary conditions" and the significance and utility of the Courant–Friedrich–Lewy (CFL) number.*

The following sentences have been added to the manuscript:

"[…] At the front and back of this section, periodic boundary conditions are used. **Periodic boundary conditions in one or more space directions imply that any fluid field is periodically continued across the domain size in this direction, e.g. the temperature field is periodic in *x* if *T(x+L,y,z,t)= T(x,y,z,t)* with the box length *L* in *x*.** […]"

"[…] it is adjusted automatically to ensure a CFL number below 0.95. **The CFL number is a parameter for the numerical solution of partial differential equations. The discrete time step width $\Delta t$ in numerical simulations has to be chosen depending on the local velocity magnitude *U* in the mesh cells and their local widths $\Delta x$ to guarantee the stability of the numerical method. In detail, it should hold that $\Delta t \leq \Delta x/U$. The CFL number is the corresponding dimensionless quantity, *C = $\Delta t U/\Delta x$*. In our case, *C* should be smaller than 1.** […]"

*Line 225: What is the relevant particle size range and fluid velocity range being considered when evaluating the particle Reynolds number?*

We have re-written this part of the manuscript including a statement about the relevant particle size range and fluid velocity:

"[…] being the particle's density and diameter. **The coefficient $C_D$ depends on particle Reynolds number $Re_P$ and is usually calculated according to Stokes (1851) for low $Re_P$, and for higher $Re_P$ according to Schiller and Naumann (1933):**

$$C_D = \begin{cases} \dfrac{24}{Re_P} & \text{for } Re_P < 0.5 \\ \dfrac{24}{Re_P}(1 + 0.15Re_P^{0.687}) & \text{for } Re_P \geq 0.5 \end{cases}. \tag{5}$$

**$Re_P$ is calculated with the current values of $D_p$ and the slip velocity $|U_f – U_p|$ at every Lagrangian time step:**

$$Re_P = \frac{D_P|U_F - U_p|}{\nu_F}, \tag{6}$$

**with $\nu_f$ being the kinematic viscosity of the fluid. As the particles/droplets are rather small ($1 \cdot 10^{-7}$ m < $D_p$ < $1 \cdot 10^{-5}$ m), they are assumed to follow the advecting flow field nearly perfectly, i.e., the slip velocity is small compared to the fluid velocity. Thus, $Re_P$ is assumed to be small.**

*$C_{LS}$ is calculated [...]*"

*Fig. 1: Please indicate the air flow direction in the various channels.*

The air-flow direction was added to the figure.

[Figure]

Figure 1. A schematic of LACIS-T including photos of individual components (© by Ingenieurbüro Mathias Lippold, VDI; TROPOS). **The red arrows indicate the flow direction.**

*References*

Chandrakar, K. K., Cantrell, W., Chang, K., Ciochetto, D., Niedermeier, D., Ovchinnikov, M., Shaw, R. A., and Yang, F.: Aerosol indirect effect from turbulence-induced broadening of cloud-droplet size distributions, Proceedings of the National Academy of Sciences, 113, 14 243–14 248, https://doi.org/10.1073/pnas.1612686 113, 2016.

Chandrakar, K. K., Saito, I., Yang, F., Cantrell, W., Gotoh, T., and Shaw, R. A.: Droplet size distributions in turbulent clouds: experimental evaluation of theoretical distributions, Quarterly Journal of the Royal Meteorological Society, pp. 1–22, https://doi.org/10.1002/qj.3692, 2019.

McGraw, R. and Liu, Y.: Brownian drift-diffusion model for evolution of droplet size distributions in turbulent clouds, Geophysical Research Letters, 33, 2006.

Randers-Pehrson, N.: Pioneer wind tunnels, Smithsonian Miscellaneous Collections, 93, 1935.

Saito, I., Gotoh, T., and Watanabe, T.: Broadening of cloud droplet size distributions by condensation in turbulence, Journal of the Meteorological Society of Japan. Ser. II, 97, 867–891, https://doi.org/10.2151/jmsj.2019–049, 2019.

Schiller, L. and Naumann, A.: Fundamental calculations in gravitational processing, Zeitschrift Des Vereines Deutscher Ingenieure, 77, 318–320, 1933.

Shi, X., Hagen, H. L., Chow, F. K., Bryan, G. H., and Street, R. L.: Large-eddy simulation of the stratocumulus-capped boundary layer with explicit filtering and reconstruction turbulence modeling, Journal of the Atmospheric Sciences, 75, 611–637, https://doi.org/10.1175/JAS–D–17–0162.1, 2018.

Siebert, H. and Shaw, R. A.: Supersaturation fluctuations during the early stage of cumulus formation, Journal of the Atmospheric Sciences, 74, 975–988, https://doi.org/10.1175/JAS–D–16–0115.1, 2017.

Stokes, G. G.: On the effect of the internal friction of fluids on the motion of pendulums, vol. 9, Pitt Press Cambridge, 1851.

Wang, L.-P. and Grabowski, W. W.: The role of air turbulence in warm rain initiation, Atmospheric Science Letters, 10, 1–8, https://doi.org/10.1002/asl.210, 2009.

---

## Author Response (AR1)

**Referee 1**

*The design and performance of a new system for studying turbulent effect on cloud microphysics under short timescales up to a few seconds is presented, the LACIS-T system. CFDC simulations using large eddy simulations are also performed to help interpret the system's performance and experimental results. This is an important contribution to the atmospheric science community given the lack of experimental systems to directly study the effects of turbulence. I recommend it for publication, and make a few suggestions to further improve the manuscript below.*

We thank referee 1 for his/her remarks/comments/suggestions. They are addressed below and we have revised the manuscript accordingly.

*While the system's design and characterization are nicely presented in great detail, it stuck me that a discussion of how finely the revenant parameters can be adjusted and controlled in LACIS-T was not really presented. This would be a valuable addition to the paper.*

The accuracy for flow rate, temperature and dew-point temperature adjustments have been added in section 2 where the corresponding devices are described (changes are given in bold here). For example, **the accuracy of the temperature adjustments** in the humidification system and heat exchangers as well as of their monitoring **is +/- (0.03°C +0.0005 x *T*) with *T* being the actual temperature (in °C)**. Note that, **the accuracy for monitoring the set dew-point with the dew-point mirror is ≤ ±0.1 K with a reproducibility of ≤ ±0.05 K**. The **Huber thermostats** used in the humidification system and connected to the heat exchangers **feature a temperature stability of +/- 0.01 K at -10°C**. The volume flow rate is monitored by means of **ultrasonic flow meters each of which features an accuracy of 1.5% of the reading**. **The relative measurement uncertainty of the Hot-wire anemometer is about 3%**.

*In the introduction, more elaboration on the importance of atmospheric turbulence and its effects on important properties and phenomena such as cloud microphysics and particle deliquescence/growth and the resulting rather complex and intriguing size distributions is warranted. This will be the results presented in better context.*

A respected paragraph has been inserted which is marked in bold:
"Turbulence drives processes such as entrainment and mixing, leading to strong fluctuations in aerosol particle concentration, temperature, water vapor, and consequently supersaturation which affects having implications for cloud droplet activation, growth and decay (Siebert et al., 2006; Siebert and Shaw, 2017; Chandrakar et al., 2016). **Indeed, it has been shown that representation of unresolved fluxes in large-eddy simulations influence properties of simulated stratocumulus clouds (Shi et al., 2018), and that the range of scales captured in direct numerical simulations of cloud entrainment influence the width of the droplet size distribution (Kumar et al., 2018). Even without the presence of strong entrainment, fluctuations in supersaturation can influence the functional form of the cloud droplet size distribution (e.g., McGraw and Liu, 2006; Chandrakar et al., 2016; Saito et al., 2019; Chandrakar et al., 2019). Turbulence also influences particle collision rates and is therefore thought to be central to precipitation formation (Shaw, 2003; Wang and Grabowski, 2009).** These processes, in turn, can have buoyancy and drag effects on turbulence and influence cloud dynamic processes up to the largest scales (Stevens et al, 2005; Malinowski et al., 2008; Bodenschatz et al., 2010)".

*Line 91: Is there a reference for the "Göttingen type" of wind tunnel?*

The Göttingen wind tunnel goes back to Prandtl and Betz. It is a closed-loop wind-tunnel with
a measurement section. A citation was added to the text: Randers-Pehrson (1935) which

60 gives a review about wind tunnels of that time. There it is written: "With the construction of
the first wind tunnel at Göttingen we are approaching modern times. This was the first return-
flow tunnel, built by Dr. Ludwig Prandtl for Motorluftschiffstudiengesellschaft and completed
in July 1908. This tunnel was superseded in 1916-17 by a much larger tunnel with
open jet and return flow, which is now called the Göttingen type."

65

*Lines 205-210: To make this more accessible to those less familiar with CFD simulations,
please explain "periodic boundary conditions" and the significance and utility of the Courant–
Friedrich–Lewy (CFL) number.*

70
The following sentences have been added to the manuscript:

"[…] At the front and back of this section, periodic boundary conditions are used. **Periodic
boundary conditions in one or more space directions imply that any fluid field is**

75 **periodically continued across the domain size in this direction, e.g. the temperature
field is periodic in *x* if *T(x+L,y,z,t)= T(x,y,z,t)* with the box length *L* in *x*.** […]"

"[…] it is adjusted automatically to ensure a CFL number below 0.95. **The CFL number is a
parameter for the numerical solution of partial differential equations. The discrete time**

80 **step width $\Delta t$ in numerical simulations has to be chosen depending on the local
velocity magnitude *U* in the mesh cells and their local widths $\Delta x$ to guarantee the
stability of the numerical method. In detail, it should hold that $\Delta t \leq \Delta x/U$. The CFL
number is the corresponding dimensionless quantity, *C* = $\Delta t U/\Delta x$. In our case, *C*
should be smaller than 1.** […]"

85

*Line 225: What is the relevant particle size range and fluid velocity range being considered
when evaluating the particle Reynolds number?*

90 We have re-written this part of the manuscript including a statement about the relevant
particle size range and fluid velocity:

"[…] being the particle's density and diameter. **The coefficient $C_D$ depends on particle
Reynolds number Re$_P$ and is usually calculated according to Stokes (1851) for low**

95 **Re$_p$, and for higher Re$_P$ according to Schiller and Naumann (1933):**

$$C_D = \begin{cases} \dfrac{24}{\mathrm{Re}_P} & \text{for } \mathrm{Re}_P < 0.5 \\ \dfrac{24}{\mathrm{Re}_P}(1+0.15\mathrm{Re}_P^{0.687}) & \text{for } \mathrm{Re}_P \geq 0.5 \end{cases}. \tag{5}$$

**Re$_P$ is calculated with the current values of $D_p$ and the slip velocity $|U_f – U_p|$ at every
Lagrangian time step:**

$$\mathbf{Re}_P = \frac{D_P|U_F-U_p|}{\nu_F}, \tag{6}$$

**with $\nu_f$ being the kinematic viscosity of the fluid. As the particles/droplets are rather**

100 **small ($1\cdot10^{-7}$ m $< D_p <$ $1\cdot10^{-5}$ m), they are assumed to follow the advecting flow field
nearly perfectly, i.e., the slip velocity is small compared to the fluid velocity. Thus, Re$_P$
is assumed to be small.**

*$C_{LS}$ is calculated [...]*"

105

*Fig. 1: Please indicate the air flow direction in the various channels.*

The air-flow direction was added to the figure.

110

[Figure]

Figure 1. A schematic of LACIS-T including photos of individual components (© by Ingenieurbüro Mathias Lippold, VDI; TROPOS). **The red arrows indicate the flow direction.**

115

155

**Referee 2**

*General Comment: This manuscript presents the newly developed turbulent moist-air wind tunnel, called the Turbulent Leipzig Aerosol Cloud Interaction Simulator (LACIST). LACIS-T is able to study different cloud processes taking into account interactions between turbulence and cloud microphysical processes. Additionally, the authors complemented their LACIS-T experiments with Computational Fluid Dynamics (CFD) simulations to explain their observations. The behavior of the LACIS-T was tested by performing deliquescence and hygroscopic growth as well as droplet activation and growth experiments using NaCl particles. This is as well written manuscript, with a very detailed descriptions of this newly developed turbulent moist-air wind tunnel. The LACIS-T is a great and valuable instrument for the cloud physics community that can be used to fulfill many gaps in knowledge. Given the lack of instruments like this, LACIS-T can have a huge impact in the near future. I congratulate the authors for developing such a great instrument and for the careful characterization. I only have one "Major Comment". The manuscript can be accepted after the following minor comments are added to the revised manuscript.*

We thank referee 2 for his/her remarks/comments/suggestions. They are addressed below and we have revised the manuscript accordingly.

*Major Comment: It would have been nice to add a reference experiment, especially for the droplet activation experiments. I mean, is it possible to run a droplet activation experiment under steady conditions, i.e., without any turbulence? This will show how monodisperse is the droplet size distribution (DSD) in comparison to the DSD shown in Figure 12.*

We agree that such a reference experiment would be very valuable. Therefore, **as kind of a benchmark, we want to evaluate how the droplet size distribution would look like without turbulence affecting droplet formation and growth. Therefore, utilizing the above described numerical model, two additional cases have been investigated, a) a case without grid induced turbulence (i.e., the grid was removed from the numerical simulation) and b) an idealized case based on time-averaged flow fields without turbulent fluctuations. In these simulations, the formation and growth of NaCl particles, with $D_{p,dry}$ = 100 nm for the temperature difference of $\Delta T$ = 16K was considered. The simulation results are shown in Fig. 13 for $z_8$ = 80cm together with measurement results, for which the turbulence grid remains included as shown in Fig. 12.**

[Figure]

***Figure 13.*** *Comparison between different model calculations for the particle formation and growth on NaCl particles with $D_{p,dry}$ = 100 nm at $z_8$ = 80cm for $\Delta T$ = 16K. Left figure: LES with turbulence grid (as shown in lower right plot of Fig. 12). Middle figure: LES but without turbulence grid. Right figure: simulation with averaged fields used as frozen flow fields, and transient particle calculation. In all plots, the measurement results, for which the turbulence grid is included (as shown in Fig.12), are shown for reference.*

**It turns out that the removal of the turbulence grid does not lead to laminar conditions. We still observe inherent turbulent conditions due to wall effects and the high Reynolds number (order of $10^4$) for the set velocity. But the turbulence intensity and therefore the strength of turbulent fluctuations is decreased. The power spectra obtained for the configuration without grid further suggest that the turbulence is anisotropic in this case (not shown). As a consequence, we still obtain a broad droplet size distribution (see middle plot in Fig. 13) which is however narrower compared to the measurement / simulation with turbulence grid. We further observe a significant number of particles close to $D_p$ = 300 nm.**
**Laminar conditions, which would lead to a very narrow droplet size distribution (see right plot in Fig. 13), can only be simulated if averaged fields are used as frozen flow fields in the simulations, which is not realizable in the real experiment. In other words, measurements without turbulence are not executable inside LACIS-T and the flow regime is best controlled in presence of the turbulence grid. Furthermore, these simulations clearly indicate the distinct influences of turbulence on the droplet size distributions, and consequently, on the formation and growth of droplets inside LACIS-T.**
The text marked in bold as well as Fig. 13 have been added at the end of section 5.2.

*Minor Comments:*
*L19: Add a reference after "Earth".*

Lamb and Verlinde (2011) has been added to the text.

*L20: Add a reference after "interactions".*

Mason and Ludlam (1951), Hobbs (1991), and Kreidenweis et al. (2019) have been added to the text.

*L24: Add a reference after "scales".*

The sentence has been connected with the followed-up sentence in order to avoid citing the same reference twice.

Atmospheric clouds are often non–stationary, inhomogeneous, intermittent, and cover an enormous range of spatial (micrometers to hundreds of kilometers) and temporal (microseconds to hours and days) scales.  with cross–scale interactions between turbulent fluid dynamics and cloud microphysical processes  influencing cloud behavior and cloud development (Bodenschatz et al., 2010).

*L28: I suggest to add other references in addition to Siebert et al. (2006).*

The sentence has been changed slightly and additional references have been added: Turbulence drives processes such as entrainment and mixing, leading to strong fluctuations in aerosol particle concentration, temperature, water vapor, and consequently supersaturation  **with implications for** cloud droplet activation, growth and decay (Siebert et al., 2006; Chandrakar et al., 2016, Siebert et al., 2017).

*L28: "It links to phase transition processes". Do the authors refer to "turbulence"?*

Yes, this is right. However, due to a comment by reviewer 1, the sentence has been

rewritten: "[…] **Turbulence also influences particle collision rates and is therefore thought to be central to precipitation formation (Shaw, 2003; Wang and Grabowski, 2009)**  […]".

265

*L34: Add a reference after "undertaking".*

Stratmann et al. (2009) has been added.

270

*L37: I suggest to add other references in addition to Stratmann et al. (2009).*

List et al. (1986) and Kreidenweis et al. (2019) have been added.

275

*L40: How about Cziczo et al. (2017)?*

The citation Cziczo et al. (2017) has been added to the text.

280

*L44-49: I do not think it is necessary to cite all this previous papers.*

We agree. Now in almost all cases, two papers are cited per particle type/species.

285

*L50: I think "those of the other" should be "those of other".*

We agree, it has been changed accordingly.

290

*L51: Add a reference after "interactions".*

Chang et al. (2016) has been added to the text.

295

*L62-73: Much of the information provided here can go into methods.*

This part has been written in order to get a first impression about the set-up and benefits of
300  LACIS-T as well as answering the question what distinguishes this wind tunnel from other
facilities like the PI chamber. Therefore, we would like to leave the main part here only
deleting the last sentence of this paragraph.

305  *L104: "to remove aerosol particles". In the particle-free air?*

The sentences have been re-written:

[revised manuscript text omitted]
_p}} & \text{for } \mathrm{Re_p} < 0.5 \\ \frac{24}{\mathrm{Re_p}} \left( 1 + 0.15 \mathrm{Re_p}^{0.687} \right) & \text{for } \mathrm{Re_p} \geq 0.5. \end{cases} \tag{5}$$

 $\mathrm{Re_p}$ **is calculated with the current values of $D_{\mathrm{p}}$ and the slip velocity $|\boldsymbol{U}_{\mathrm{f}} - \boldsymbol{U}_{\mathrm{p}}|$ at every Lagrangian time step:**

$$\mathrm{Re_p} = \frac{D_{\mathrm{p}} |\boldsymbol{U}_{\mathrm{f}} - \boldsymbol{U}_{\mathrm{p}}|}{\nu_{\mathrm{f}}}, \tag{6}$$

with $\nu_f$ being the kinematic viscosity of the fluid. **As the particles/droplets are rather small ($1\times10^{-7}$ m $< D_p < 1\times10^{-5}$ m), they are assumed to follow the advecting flow field nearly perfectly, i.e., the slip velocity is small compared to the fluid velocity. Thus, $\mathrm{Re}_p$ is assumed to be small.**

255   $C_{LS}$ is calculated according to Mei (1992):

$$C_{LS} = \frac{4.1126}{\mathrm{Re_s}^{0.5}} f(\mathrm{Re_p}, \mathrm{Re_s}), \tag{7}$$

with

$$f(\mathrm{Re_p}, \mathrm{Re_s}) = \begin{cases} \left(1 - 0.3314\beta^{0.5}\right)\exp\left(-\frac{\mathrm{Re_p}}{10}\right) + 0.3314\beta^{0.5} & \text{for } \mathrm{Re_p} < 40 \\ 0.0524\beta\mathrm{Re_p} & \text{for } \mathrm{Re_
[revised manuscript text omitted]
_\mathrm{d} = 0\,\mathrm{K}$; moist with $\Delta T_\mathrm{d} = 0\,\mathrm{K}$; and moist with $\Delta T_\mathrm{d} = 4\,\mathrm{K}$, exemplarily for $z_3$. As expected, the temperature profiles shown in the left of Fig. 9 exhibit a very similar behavior, i.e., the influence of the increased amount of water vapor in the air

385 flow as well as the water vapor profile itself (in terms of RH) on the temperature curve is very low. The reasons are that still less than 1% of the total mass is water vapor which does not significantly influence the fluid properties (e.g., heat capacity). Further, there is no condensation of water vapor which could influence the temperature profile due to latent heat release.

On the right hand side of Fig. 9 the temperature and water vapor mixing are shown for the moist case ($\Delta T_\mathrm{d} = 4\,\mathrm{K}$). In order to compare both quantities, the normalized water vapor mixing ratio $\xi_\mathrm{n}$ and the normalized temperature $\theta_\mathrm{n}$ are depicted.

390 The normalized water vapor mixing ratio is defined as $\xi_\mathrm{n} = (q_\mathrm{v} - q_{\mathrm{v},1})/(q_{\mathrm{v},2} - q_{\mathrm{v},1})$, where $q_{\mathrm{v},1}$ and $q_{\mathrm{
[revised manuscript text omitted]